# Neural MJD: Neural Non-Stationary Merton Jump Diffusion for Time Series Prediction

**Yuanpei Gao**[1,2]
yuanpeig@student.ubc.ca

**Qi Yan**[1,2]
qi.yan@ece.ubc.ca

**Yan Leng**[3]
yan.leng@mccombs.utexas.edu

**Renjie Liao**[1,2,4]
rjliao@ece.ubc.ca

[1]University of British Columbia; [2]Vector Institute
[3]University of Texas at Austin; [4]Canada CIFAR AI Chair

## Abstract

While deep learning methods have achieved strong performance in time series prediction, their black-box nature and inability to explicitly model underlying stochastic processes often limit their robustness handling non-stationary data, especially in the presence of abrupt changes. In this work, we introduce *Neural MJD*, a neural network based non-stationary Merton jump diffusion (MJD) model. Our model explicitly formulates forecasting as a stochastic differential equation (SDE) simulation problem, combining a time-inhomogeneous Itô diffusion to capture non-stationary stochastic dynamics with a time-inhomogeneous compound Poisson process to model abrupt jumps. To enable tractable learning, we introduce a likelihood truncation mechanism that caps the number of jumps within small time intervals and provide a theoretical error bound for this approximation. Additionally, we propose an Euler-Maruyama with restart solver, which achieves a provably lower error bound in estimating expected states and reduced variance compared to the standard solver. Experiments on both synthetic and real-world datasets demonstrate that Neural MJD consistently outperforms state-of-the-art deep learning and statistical learning methods. Our code is available at https://github.com/DSL-Lab/neural-MJD.

## 1 Introduction

Real-world time series often exhibit a mix of continuous trends and abrupt changes (jumps) [1, 2]. For example, stock prices generally follow steady patterns driven by macroeconomic factors but can experience sudden jumps due to unexpected news or policy shifts [3, 4]. Similarly, retail revenue may rise seasonally but jump abruptly due to sales promotions or supply chain disruptions [5, 6]. These discontinuous changes pose significant challenges for temporal dynamics modeling.

Classical statistical models, *e.g.*, Merton jump diffusion (MJD) [3] or more general Lévy processes [7], provide a principled approach for modeling such data with jumps. They are effective for small datasets with well-understood statistical properties [8–10]. However, their assumptions—such as independent and stationary increments—often fail in real-world non-stationary settings. Additionally, these models struggle to capture interdependencies across multiple time series, such as competition effects among colocated businesses [11] or spillover dynamics in stock markets driven by investor attention shifts [12, 13]. As a result, they are difficult to scale effectively to large datasets. In contrast, deep learning approaches have demonstrated strong empirical performance by effectively learning time-varying patterns from data [14–19]. Despite their success, these models are often black-box in nature and lack explicit mathematical formulations to describe the underlying dynamics. This limits their

39th Conference on Neural Information Processing Systems (NeurIPS 2025).

interpretability and often results in poor generalization to non-stationary data with jumps. Notably, in previous deep learning time-series studies, "non-stationarity" typically refers to distributional shifts in the data over time. These works focus on mitigating such shifts using techniques like input-level normalization (*e.g.*, DAIN [20], ST-norm [21], RevIN [22]), or domain adaptation (*e.g.*, DDG-DA [23]). In contrast, our notion of "non-stationarity" centers on modeling a MJD process with parameters that evolve over time.

To address these limitations, we propose *Neural MJD*, a neural parameterization of the non-stationary Merton jump diffusion model that combines the advantages of statistical and learning-based approaches. In particular, our contributions are as follows:

- Our *Neural MJD* integrates a time-inhomogeneous Itô diffusion to capture non-stationary stochastic dynamics and a time-inhomogeneous compound Poisson process to model abrupt jumps. The parameters of the corresponding SDEs are predicted by a neural network conditioned on past data and contextual information.
- To enable tractable learning, we present a likelihood truncation mechanism that caps the number of jumps within small time intervals and provide a theoretical error bound for this approximation. Additionally, we propose an *Euler-Maruyama with restart* solver for inference, which achieves a provably lower error bound in estimating expected states and reduced variance compared to the standard solver.
- Extensive experiments on both synthetic and real-world datasets show that our model consistently outperforms deep learning and statistical baselines under both stochastic and deterministic evaluation protocols.

## 2 Related work

Many neural sequence models have been explored for time series prediction, *e.g.*, long short-term memory (LSTM) [24], transformers [25], and state space models (SSMs) [26]. These models [27–36] have shown success across domains, including industrial production [37], disease prevention [38, 39], and financial forecasting [40, 31]. To handle more contextual information, extensions incorporating spatial context have been proposed, *e.g.*, spatio-temporal graph convolutional networks (STGCN) [41], diffusion convolutional recurrent neural networks (DCRNN) [42], and graph message passing networks (GMSDR) [43]. However, these models remain fundamentally deterministic and do not explicitly model stochastic temporal dynamics. Generative models, *e.g.*, deep auto-regressive models [44] and diffusion/flow matching models [45–47], provide probabilistic modeling of the time series and generate diverse future scenarios [48–53]. However, these models often face computational challenges, as either sampling or computing the likelihood can be expensive. Additionally, they do not explicitly model abrupt jumps, limiting their generalization ability to scenarios with discontinuities.

Another line of research integrates classical mathematical models, such as ordinary and stochastic differential equations (ODEs and SDEs), into deep learning frameworks [33, 54–59]. In financial modeling, physics-informed neural networks (PINNs) [60] have been explored to incorporate hand-crafted Black-Scholes (BS) and MJD models as guidance to construct additional loss functions [61, 62]. However, these approaches differ from ours, as we directly parameterize the non-stationary MJD model using neural networks rather than imposing predefined model structures as constraints. Neural jump diffusion models have also been explored in the context of temporal point processes (TPPs), such as Hawkes and Poisson processes [63, 64]. However, these methods primarily focus on event-based modeling, where jumps are treated as discrete occurrences of events, thus requiring annotated jump labels during training. In contrast, our approach aims to predict time series values at any give time, irrespective of whether a jump occurs, without relying on labeled jump events. Moreover, since jump events are unknown in our setting, our likelihood computation is more challenging since it requires summing over all possible number of jumps.

Finally, various extensions of traditional MJD have been proposed in financial mathematics to handle non-stationary data [65, 66], such as the stochastic volatility jump (SVJ) model [67], affine jump models [68], and the Kou jump diffusion model [69]. However, these models rely on strong assumptions for analytical tractability, requiring manual design of parameter evolution and often being computationally expensive [70]. For example, the SVJ model combines Heston's stochastic volatility with MJD under the assumption that volatility follows the Cox-Ingersoll-Ross (CIR) process, meaning it reverts to a fixed long-term mean. Despite this, it lacks a closed-form likelihood function. Moreover,

variants with time-dependent parameters require calibrations on market data to obtain the functions of parameters [71]. In contrast, our model directly learns the parameters of the non-stationary MJD from data, which is similar to the classical SDE calibration for financial modeling, but provides better expressiveness and flexibility while still permitting closed-form likelihood evaluation.

## 3 Background

To better explain our method, we first introduce two prominent models in mathematical finance, the Black-Scholes model and the Merton jump diffusion model.

**Black-Scholes (BS) model.** The Black-Scholes model was developed by Fischer Black and Myron Scholes, assuming that asset prices follow a continuous stochastic process [72]. Specifically, the dynamics of asset price $S_t$ at time $t$ is described by the following SDE:

$$\mathrm{d}S_t = S_t(\mu \mathrm{d}t + \sigma \mathrm{d}W_t), \tag{1}$$

where $\mu$ is the drift rate, representing the expected return per unit of time, and $\sigma$ is the volatility, indicating the magnitude of fluctuations in the asset price. $W_t$ refers to a standard Wiener process.

**Merton jump diffusion (MJD) model.** To account for discontinuities in asset price dynamics, Robert C. Merton extended the BS model by introducing the MJD model [3]. This model incorporates an additional jump process that captures sudden and significant changes in asset prices, which cannot be explained by continuous stochastic processes alone.

The dynamics of the asset price $S_t$ in the MJD model are described by the following SDE:

$$\mathrm{d}S_t = S_t((\mu - \lambda k)\mathrm{d}t + \sigma \mathrm{d}W_t + \mathrm{d}Q_t), \tag{2}$$

where $Q_t$ follows a compound Poisson process and captures the jump part. Specifically, $Q_t = \sum_{i=1}^{N_t}(Y_i - 1)$, where $Y_i$ is the price ratio caused by the $i$-th jump event occurring at the time $t_i$, *i.e.*, $Y_i = S_{t_i}/S_{t_i^-}$ and $N_t$ is the total number of jumps up to time $t$. $S_{t_i}$ and $S_{t_i^-}$ are the prices after and before the jump at time $t_i$, respectively. $Y_i$ captures the relative price jump size since $\mathrm{d}S_{t_i}/S_{t_i} = (S_{t_i} - S_{t_i^-})/S_{t_i^-} = Y_i - 1$. The price ratio $Y_i$ follows a log-normal distribution, *i.e.*, $\ln Y_i \sim \mathcal{N}(\nu, \gamma^2)$, where $\nu$ and $\gamma^2$ are the mean and the variance respectively. $N_t$ denotes the number of jumps that occur up to time $t$ and follows a Poisson process with intensity $\lambda$, which is the expected number of jumps per unit time. To make the expected relative price change $\mathbb{E}[\mathrm{d}S_{t_i}/S_{t_i}]$ remain the same as in the BS model in Eq. (1), MJD introduces an additional adjustment in the drift term of the diffusion, *i.e.*, $-\lambda k\mathrm{d}t$ in Eq. (2). In particular, we have,

$$\mathbb{E}[\mathrm{d}Q_t] = \mathbb{E}[(Y_{N_t} - 1)\mathrm{d}N_t] = \mathbb{E}[Y_{N_t} - 1]\mathbb{E}[\mathrm{d}N_t],$$

where we use the assumption of MJD that "how much it jumps" (captured by $Y_{N_t}$) and "when it jumps" (captured by $N_t$) are independent. For the log-normal distributed $Y_{N_t}$, we can compute the expected jump magnitude $\mathbb{E}[Y_{N_t} - 1] = \exp{(\nu + \gamma^2/2)} - 1$. To simplify the notation, we define $k := \mathbb{E}[Y_{N_t} - 1]$. For the Poisson process $N_t$, we have $\mathbb{E}[\mathrm{d}N_t] = \lambda \mathrm{d}t$. Therefore, we have $\mathbb{E}[\mathrm{d}Q_t] = \lambda k\mathrm{d}t$, which justifies the adjustment term $-\lambda k\mathrm{d}t$ in Eq. (2).

The MJD model has an explicit solution for its asset price dynamics, given by:

$$\ln \frac{S_t}{S_0} = \left(\mu - \lambda k - \frac{\sigma^2}{2}\right)t + \sigma W_t + \sum_{i=1}^{N_t} \ln Y_i. \tag{3}$$

Based on this solution, the conditional probability of the log-return at time $t$, given the initial price $S_0$ and the number of jumps $N_t = n$, can be derived as:

$$P(\ln S_t | S_0, N_t = n) = \mathcal{N}\left(a_n, b_n^2\right), \tag{4}$$

where $a_n = \ln S_0 + \left(\mu - \lambda k - \frac{\sigma^2}{2}\right)t + n\nu$ and $b_n^2 = \sigma^2 t + \gamma^2 n$. Therefore, we obtain the likelihood,

$$P(\ln S_t | S_0) = \sum_{n=0}^{\infty} P(N_t = n) P(\ln S_t | S_0, N_t = n) = \sum_{n=0}^{\infty} \frac{(\lambda t)^n}{n!} \frac{1}{\sqrt{2\pi b_n^2}} \exp\left(-\frac{(\ln S_t - a_n)^2}{2b_n^2}\right). \tag{5}$$

Here we use the fact that $P(N_t = n)$ follows a Poisson distribution. One can then perform the maximum likelihood estimation (MLE) to learn the parameters $\{\mu, \sigma, \lambda, \nu, \gamma\}$. Additionally, the conditional expectation has a closed-form,

$$\mathbb{E}[S_t | S_0] = S_0 \exp(\mu t). \tag{6}$$

The derivations of the above formulas are provided in App. A.

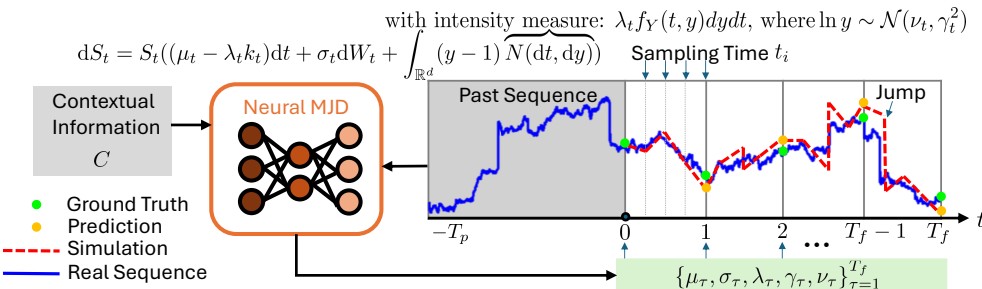

Figure 1: The overview of Neural MJD. Our model captures discontinuous jumps in time-series data and uncovers the underlying non-stationary SDEs from historical sequences and context information. Our method enables numerical simulations for future forecasting along time.

## 4 Methods

We consider the general time series prediction problem where, given the observed past data $\{S_0, S_{-1}, \ldots, S_{-T_p}\}$, and optional contextual information (*e.g.*, additional features) $C$, the goal is to predict the future values $\{S_1, \ldots, S_{T_f}\}$. Here $T_p$ and $T_f$ denote the past and future time horizons, respectively. While we use integer indices for denoting time, sampling time is not restricted to integers. Our model is built upon a diffusion process, which is inherently continuous in time and compatible with arbitrary sampling mechanism. An overview of our method is shown in Fig. 1.

### 4.1 Neural Non-Stationary Merton Jump Diffusion

In the vanilla MJD model, the increments from the Wiener and Compound Poisson processes are independent and stationary, *i.e.*, $\sigma dW_t \overset{\text{iid}}{\sim} \mathcal{N}(0, \sigma^2 dt)$, $dN_t \overset{\text{iid}}{\sim} \text{Pois}(\lambda dt)$, and $\ln Y_i \overset{\text{iid}}{\sim} \mathcal{N}(\nu, \gamma^2)$. The stationary assumption may be too strong in the real-world applications. For example, the stock prices of certain companies, such as Nvidia, experienced significantly larger jumps in the past decade compared to the previous one. Specifically, we allow independent but non-stationary increments in MJD by introducing time-inhomogeneous parameters $\{\mu_t, \sigma_t, \lambda_t, \gamma_t, \nu_t\}_{0 \leq t \leq T_f}$ in the original SDE. Thus, at any future time $t$, the modified SDE follows

$$dS_t = S_t \left( (\mu_t - \lambda_t k_t) dt + \sigma_t dW_t + \int_{\mathbb{R}^d} (y - 1) N(dt, dy) \right). \tag{7}$$

Here $\lambda_t \in \mathbb{R}_+$ and $\sigma_t, \mu_t, k_t \in \mathbb{R}^d$, while $W_t$ denotes a $d$-dimensional standard Wiener process. With a slight abuse of notation, $\sigma_t dW_t$ means element-wise product between two size-$d$ vectors. $N(dt, dy)$ is a *Poisson random measure* on $[0, T_f] \times \mathbb{R}^d$, which encodes both the timing and magnitude of jumps. Intuitively, a Poisson random measure extends the idea of a Poisson process to random events distributed over both time and magnitude spaces, and $N(dt, dy)$ counts the number of jumps occurring within the infinitesimal time interval $[t, t + dt]$ whose sizes fall within $[y, y + dy]$. The jump component $S_t \int_{\mathbb{R}^d} (y - 1) N(dt, dy)$ introduces abrupt discontinuities in the process, accounting for sudden shifts in data.

The statistical properties of the Poisson random measure are uniquely determined by its intensity measure $\lambda_t f_Y(t, y) dy dt$. In our model, the intensity measure controls time-inhomogeneous jump magnitudes and frequencies. Namely, jump times follow a Poisson process with time-dependent intensity $\lambda_t$ and jump magnitudes follow a time-dependent log-normal distribution, *i.e.*, a jump $Y_t$ at time $t$ follows $\ln Y_t \sim \mathcal{N}(\nu_t, \gamma_t^2)$, where we denote the log-normal density of $Y_t$ as $f_Y(t, y)$. Similarly, we define $k_t := \mathbb{E}[Y_t - 1] = \exp(\nu_t + \frac{\gamma_t^2}{2}) - 1$ in the drift term. This makes Eq. (7) equivalent to using the compensated Poisson measure $\tilde{N}(dt, dy) := N(dt, dy) - \lambda_t f_Y(t, y) dy dt$ to remove the expected contribution of jumps. Note Eq. (7) includes $k_t$, so that $\int_{\mathbb{R}^d} (y - 1) \lambda_t f_Y(t, y) dy dt = \lambda_t \mathbb{E}[Y_t - 1] dt = \lambda_t k_t dt$. Namely, it can be rewritten as,

$$dS_t = S_t \left( \mu_t dt + \sigma_t dW_t + \int_{\mathbb{R}^d} (y - 1) \tilde{N}(dt, dy) \right).$$

This preserves the martingale property of the process induced by $\tilde{N}(dt, dy)$, *e.g.*, $\mathbb{E}[dS_t/S_t]$ matches the drift term in the non-stationary Black–Scholes model without jumps.

More importantly, inspired by the amortized inference in VAEs [73], we use neural networks to predict these time-inhomogeneous parameters based on the historical data and the contextual information $C$,

$$\mu_t, \sigma_t, \lambda_t, \gamma_t, \nu_t = f_\theta(S_0, S_{-1}, \ldots, S_{-T_p}, C, t), \tag{8}$$

where $f$ is a neural network parameterized by $\theta$. To simplify the notation, we denote the set of all observed data as $\mathcal{C} := \{S_0, S_{-1}, \ldots, S_{-T_p}, C\}$ from now on. Importantly, we only train a single network across all series and optimize the conditional log-likelihood as the training objective, which differs from standard statistical inference. At test time, the network produces context-dependent estimates $(\mu_t, \sigma_t, \lambda_t, \nu_t, \gamma_t)$ across all future time in a single forward pass.

The stochastic process described by the SDE in Eq. (7) is formally an instance of an *additive process* [65, Ch. 14], characterized by independent but non-stationary increments. If $\sigma_t, \mu_t \in L^2$, *i.e.*, they are square-integrable functions, and $\max_\tau(\int_{\mathbb{R}^d} |y|^2 |\lambda_\tau f_Y(\tau, y) \mathrm{d}y) < \infty$, then our non-stationary MJD has a unique solution for every $S_0 > 0$ [65, Theorem 14.1]. As our prediction time horizon is a closed domain $t \in [0, T_f]$, these conditions are easily satisfied as long as the neural network $f_\theta$ does not produce unbounded values. At any future time $T$, the explicit solution of the SDE is given by,

$$\ln \frac{S_T}{S_0} = \int_0^T (\mu_t - \lambda_t k_t - \frac{\sigma_t^2}{2}) \mathrm{d}t + \int_0^T \sigma_t \mathrm{d}W_t + \int_0^T \int_{\mathbb{R}^d} \ln y N(\mathrm{d}t, \mathrm{d}y). \tag{9}$$

Next, we model the conditional probability of log-return $\ln S_t$ given initial price $S_0$ and past data $\mathcal{C}$,

$$P(\ln S_T | S_0, \mathcal{C}) = \sum_{n=0}^\infty \frac{\exp(-\int_0^T \lambda_t \mathrm{d}t)}{n!} \Phi_n, \tag{10}$$

where

$$\Phi_n = \int \cdots \int_{[0,T]} \Pi_{i=1}^n \lambda_{t_i} \phi(\ln S_T; a_n, b_n^2) \, \mathrm{d}t_1 \cdots \mathrm{d}t_n,$$

$$a_n = \ln S_T + \int_0^T \left( \mu_t - \lambda_t k_t - \frac{\sigma_t^2}{2} \right) \mathrm{d}t + \sum_{i=1}^n \nu_{t_i}, \; b_n^2 = \int_0^T \sigma_t^2 \mathrm{d}t + \sum_{i=1}^n \gamma_{t_i}^2.$$

Here $\phi(\ln S_T; a_n, b_n^2)$ is the density of a normal distribution with mean $a_n$ and variance $b_n^2$. $t_{1:n}$ denote the timing of $n$ jumps. Further, we compute the conditional expectations as,

$$\mathbb{E}[S_T | \mathcal{C}] = S_0 \exp(\int_0^T \mu_t \mathrm{d}t). \tag{11}$$

Please refer to App. B for derivations. Evaluating Eq. (9) and Eq. (10) is non-trivial due to time inhomogeneity and jumps, typically requiring Monte Carlo methods or partial integro-differential equation techniques for approximate solutions [65, Ch. 6, Ch. 12].

## 4.2 Tractable Learning Method

While Eq. (10) provides the exact likelihood function, evaluating it precisely is impractical due to 1) integrals with time-dependent parameters lacking closed-form solutions and 2) the infinite series over the number of jumps. To learn the model via the maximum likelihood principle, we propose a computationally tractable learning objective with parameter bootstrapping and series truncation.

First, given a finite number of future time steps $\{1, \ldots, T_f\}$, we discretize the continuous time in SDEs to construct a piecewise non-stationary MJD. Our model predicts time-varying parameters $\{\mu_\tau, \sigma_\tau, \lambda_\tau, \gamma_\tau, \nu_\tau\}_{\tau=1}^{T_f}$. For any time $t \leq T_f$, we map it to an integer index via $\rho_t := \lfloor t \rfloor + 1$. Thus, the likelihood of the data at $t + \delta$ given the data at $t$, where $\rho_t - 1 \leq t < t + \delta < \rho_t$, is given by:

$$P(\ln S_{t+\delta} | S_t, \mathcal{C}) = \sum_{n=0}^\infty P(\Delta N = n) P(\ln S_{t+\delta} | S_t, \mathcal{C}, \Delta N = n)$$

$$= \sum_{n=0}^\infty \exp(-\lambda_{\rho_t} \delta) \lambda_{\rho_t}^n \frac{\delta^n}{n!} \phi(\ln S_{t+\delta}; a_{n,\delta}, b_{n,\delta}^2), \tag{12}$$

| **Algorithm 1** Neural MJD Training | **Algorithm 2** Euler-Maruyama with Restart Inference |
|---|---|

**Algorithm 1** Neural MJD Training

1: **repeat**
2:   $(\mathcal{C}, S_{1:T_f}) \sim \mathcal{D}_{\text{train}},$
      with $\mathcal{C} = [S_{-T_p:0}, C]$
3:   $\{\mu_\tau, \sigma_\tau, \lambda_\tau, \nu_\tau, \gamma_\tau\}_{\tau=1}^{T_f} \leftarrow f_\theta(\mathcal{C})$
4:   $\hat{S}_0 \leftarrow S_0$
5:   **for** $\tau = 1, \cdots, T_f$ **do**
6:     $\psi_\tau \leftarrow \ln P(\ln S_\tau \mid S_{\tau-1} = \hat{S}_{\tau-1}, \mathcal{C})$
          $\triangleright$ Eq. (12)
7:     $\hat{S}_\tau \leftarrow \mathbb{E}[S_\tau \mid \mathcal{C}]$         $\triangleright$ Eq. (13)
8:     Update $\theta$ via
        $-\nabla_\theta \sum_{\tau=1}^{T_f} \left( -\psi_\tau + \omega \|S_\tau - \hat{S}_\tau\|^2 \right)$
9: **until** converged

**Algorithm 2** Euler-Maruyama with Restart Inference

**Require:** Solver step size $1/M$
1: $\mathcal{C} \sim \mathcal{D}_{\text{test}}$, with $\mathcal{C} = [S_{-T_p:0}, C]$
2: $\{\mu_\tau, \sigma_\tau, \lambda_\tau, \nu_\tau, \gamma_\tau\}_{\tau=1}^{T_f} \leftarrow f_\theta(\mathcal{C})$
3: $t_0 \leftarrow 0, N \leftarrow M \times T_f$
4: **for** $i = 1, \cdots, N$ **do**
5:   $t_i \leftarrow t_{i-1} + 1/M, \rho_{t_i} \leftarrow \lfloor t_i \rfloor + 1$
6:   $\alpha_i \leftarrow (\mu_{\rho_{t_i}} - \lambda_{\rho_{t_i}} k_{\rho_{t_i}} - \sigma_{\rho_{t_i}}^2/2)/M$    $\triangleright$ Drift
7:   $\beta_i \leftarrow \sigma_{\rho_{t_i}} z_1/\sqrt{M}$, with $z_1 \sim \mathcal{N}(0,1)$   $\triangleright$ Diffusion
8:   $\zeta_i \leftarrow \kappa\nu_{\rho_{t_i}} + \sqrt{\kappa}\gamma_{\rho_{t_i}} z_2$
       with $\kappa \sim \text{Pois}(\lambda_{\rho_{t_i}}/M), z_2 \sim \mathcal{N}(0,1)$    $\triangleright$ Jump
9:   **if** $(i - 1) \bmod M = 0$ **then**
10:     $\ln \bar{S}_{t_i} \leftarrow \mathbb{E}[\ln S_{\rho_{t_{i-1}}} \mid \mathcal{C}] + \alpha_i + \beta_i + \zeta_i$ $\triangleright$ Restart
11:   **else**
12:     $\ln \bar{S}_{t_i} \leftarrow \ln \bar{S}_{t_{i-1}} + \alpha_i + \beta_i + \zeta_i$
13: **return** $\{\bar{S}_{t_i}\}_{i=1}^{N}$

where $a_{n,\delta} = \ln S_t + (\mu_{\rho_t} - \lambda_{\rho_t} k_{\rho_t} - \sigma_{\rho_t}^2/2)\delta + n\nu_{\rho_t}$ and $b_{n,\delta}^2 = \sigma_{\rho_t}^2 \delta + \gamma_{\rho_t}^2 n$. This approach eliminates the need for numerical simulation to compute the integrals in Eq. (10) and has been widely adopted for jump process modeling [66, 74].

As for the conditional expectation, we have

$$\mathbb{E}\left[ S_t | \mathcal{C} \right] = S_0 \exp\left( \sum_{j=1}^{\rho_t - 1} \mu_j + (t - \rho_t + 1)\mu_{\rho_t} \right). \tag{13}$$

Derivation details are shown in App. B. Further, we jointly consider the likelihood of all future data:

$$P(\ln S_1, \cdots, \ln S_{T_f} | \mathcal{C}) = \prod_{\tau=1}^{T_f} P(\ln S_\tau | \{\ln S_j\}_{j<\tau}, \mathcal{C}) = \prod_{\tau=1}^{T_f} P(\ln S_\tau | S_{\tau-1}, \mathcal{C}),$$

where we use the Markov property and the fact that $\ln(\cdot)$ is bijective.

Therefore, the MLE objective is given by:

$$\ln P(\ln S_1, \ldots, \ln S_{T_f} \mid \mathcal{C}) = \sum_{\tau=1}^{T_f} \ln \underbrace{P\left(\ln S_\tau \mid S_{\tau-1}, \mathcal{C}\right)}_{\text{Eq. (12)}}. \tag{14}$$

The training algorithm of our neural non-stationary MJD model is shown in Alg. 1. In computing the term $\ln P(\ln S_\tau | S_{\tau-1}, \mathcal{C})$ of Eq. (14), instead of doing teacher forcing, we replace the ground truth $S_{\tau-1}$ with the conditional mean prediction $\mathbb{E}[S_{\tau-1} \mid \mathcal{C}]$ from Eq. (13). This design mitigates the discrepancy between training and inference while reducing error accumulation in sequential predictions, especially for non-stationary data. As shown in the ablation study in Sec. 5.3, this approach improves performance effectively. To further improve accuracy, we introduce an additional regularization term that encourages the conditional mean to remain close to the ground truth. Additionally, the *for* loop in Alg. 1 can be executed in parallel, as the conditional mean computation does not depend on sequential steps, significantly improving efficiency. Notably, our model imposes no restrictions on the neural network architecture, and the specific design details are provided in App. D.1.

**Truncation error of likelihood function.** Exact computation of $P(\ln S_\tau \mid S_{\tau-1}, \mathcal{C})$ in Eq. (14) requires evaluating an infinite series, which is infeasible in practice. To address this, we truncate the series at order $\kappa \in \mathbb{N}_+$, *i.e.*, limiting the maximum number of jumps between consecutive time steps. We establish the following theoretical result to characterize the decay rate of the truncation error:

**Theorem 4.1.** *Let the likelihood approximation error in Eq. (12), truncated to at most $\kappa$ jumps, be*

$$\Psi_\kappa(t, \delta) := \sum_{n=\kappa+1}^{\infty} P(\Delta N = n) P(\ln S_{t+\delta} \mid S_t, \mathcal{C}, \Delta N = n).$$

*Then, $\Psi_\kappa(t, \delta)$ decays at least super-exponentially as $\kappa \to \infty$, with a convergence rate of $O(\kappa^{-\kappa})$.*

The proof is provided in App. C.1. The truncation error is dominated by $\kappa$, with other time-dependent parameters absorbed into the big-$O$ notation. We set $\kappa$ to 5 to achieve better empirical performance.

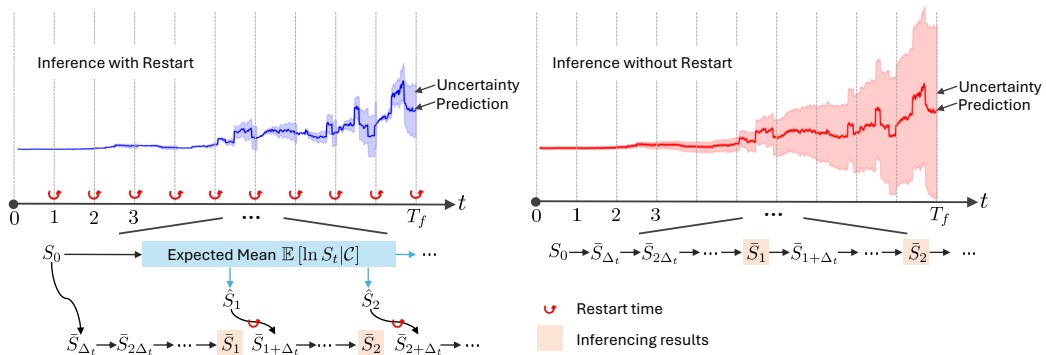

Figure 2: Comparison of numerical simulations with and without restart strategy during inference.

### 4.3 Inference based on Euler Scheme

Once trained, our Neural MJD model enables simulations following Eq. (7) by computing the non-stationary SDE parameters with a single neural function evaluation (NFE) of $f_\theta$. Unlike models limited to point-wise predictions, Neural MJD supports continuous-time simulation across the entire future horizon. Although the training data consists of a finite set of time steps, $S_{1:T_f}$, the learned model can generate full trajectories from $t = 0$ to $t = T_f$ at arbitrary resolutions.

The standard Euler-Maruyama (EM) method provides a general-purpose approach for simulating SDEs with simple implementation and proven convergence [75, 76]. However, MJD SDEs exhibit analytically derived variance that grows over time (see App. B), and the simulated trajectories produced using the vanilla EM, assuming sufficiently small error, reflect this growth as well. Notably, the resulting high-variance simulations can undermine the empirical reliability of future forecasts.

In our MJD model, it is possible to compute closed-form expressions for statistical properties such as the mean and variance at any point in time [65]; for instance, the analytical mean can be derived from Eq. (13). Building on this insight, we propose a hybrid analytic-simulation solver, the *Euler-Maruyama with restart* method, which periodically injects the exact analytical mean to improve accuracy and enhance stability, as shown in Alg. 2 and Fig. 2. Specifically, we discretize time using a uniform step size $1/M$ for simulation and set the restart points as the target times $\{1, \cdots, T_f\}$. The solver follows the standard EM method for Eq. (7) whenever a restart is unnecessary. Otherwise, it resets the state using the conditional expectation from Eq. (13).

Further, we prove that this restart strategy has a tighter weak-convergence error, particularly helpful for empirical forecasting tasks where the mean estimation is critical. Let $\epsilon_t := |\mathbb{E}[g(\bar{S}_t)] - \mathbb{E}[g(S_t)]|$ be the standard weak convergence error [75], where $S_t$ is the ground truth state, $\bar{S}_t$ is the estimated one using certain sampling scheme and $g$ is a $K$-Lipschitz continuous function. We denote the weak convergence errors of our restarted solver and the standard EM solver by $\epsilon_t^R$ and $\epsilon_t^E$, respectively.

**Proposition 4.2.** *Let $1/M$ be the step size. Both standard EM and our solver exhibit a weak convergence rate of $O(1/M)$. Specifically, the vanilla EM has a weak error of $\epsilon_t^E \le K \exp(Lt)/M$ for some constant $L > 0$, while ours achieves a tighter weak error of $\epsilon_t^R \le K \exp(L(t - \lfloor t \rfloor))/M$.*

The proof and details are left to App. C.2. Our sampler is in the same spirit as the Parareal simulation algorithms [77–80]: it first obtains estimates at discrete steps and then runs fine-grained simulations for each interval. By resetting the state to the true conditional mean at the start of each interval, our sampler reduces mean estimation error and prevents error accumulation over time. Notably, the SDE simulation requires no additional NFEs and adds negligible computational overhead compared to neural-network inference, since it involves only simple arithmetic operations that can be executed efficiently on CPUs. For reference, we also present the standard EM solver in App. D.5.

## 5 Experiments

In this section, we extensively examine Neural MJD's performance on synthetic and real-world time-series datasets, highlighting its applicability in business analytics and stock price prediction.

Table 1: Quantitative results on the **synthetic** dataset.

| Model | Mean | | Winner-takes-all | | Probabilistic | |
|---|---|---|---|---|---|---|
| | MAE↓ | $R^2$↑ | minMAE↓ | maxR²↑ | $p$-MAE↓ | $p$-R²↑ |
| ARIMA | 0.29 | –0.15 | N/A | | N/A | |
| BS | 0.25 | 0.02 | 0.20 | 0.12 | 0.22 | 0.08 |
| MJD | 0.21 | 0.08 | 0.18 | 0.15 | 0.20 | 0.09 |
| XGBoost | 0.18 | 0.17 | N/A | | N/A | |
| MLP | 0.14 | 0.21 | N/A | | N/A | |
| NJ-ODE | 0.15 | 0.20 | N/A | | N/A | |
| Neural BS | 0.15 | 0.25 | 0.10 | 0.35 | 0.14 | 0.29 |
| Neural MJD (ours) | **0.09** | **0.32** | **0.07** | **0.39** | **0.09** | **0.34** |

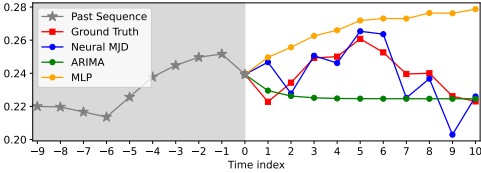

Figure 3: Qualitative result on the **synthetic** dataset.

**Baselines.** We evaluate Neural MJD against a wide range of competitors, including statistical methods such as ARIMA [81], the BS model, and the MJD model. Additionally, we compare against learning-based approaches, including supervised learning models such as XGBoost [82], MLPs, GCNs [83], as well as denoising diffusion models like DDPM [45], EDM [84], and flow matching (FM) [47]. We include recent neural ODE or SDE based learning methods such as NeuralCDE [33] and LatentSDE [59] for comparisons. NJ-ODE [85] was further included on the S&P 500 dataset. We also design a baseline model, *Neural BS*, which shares the same architecture as Neural MJD but omits the jump component. For DDPM, EDM, FM, Neural BS, and Neural MJD, we share the same transformer-based backbone to ensure a fair comparison. Since some datasets contain graph-structured data as seen in the following section, we incorporate additional graph encoding steps based on Graphormer [86] to capture spatial features, which also justifies the inclusion of GCN as a baseline. Further details are provided in App. D.

**Evaluation metrics.** We employ Mean Absolute Error (MAE), Mean Squared Error (MSE), and the R-squared ($R^2$) score as the primary evaluation metrics. To account for stochastic predictions, we run each stochastic models 10 times and report results across three types of metrics: 1) **Mean Metrics**: used for deterministic models or to average the results of stochastic models; 2) **Winner-Takes-All Metrics**: we select the best prediction from ten stochastic samples to compute minMAE, minMSE, and maxR²; 3) **Probabilistic Metrics**: these metrics assess the likelihood of stochastic predictions and select the most probable outcome to calculate $p$-MAE, $p$-MSE, and $p$-R². We mark N/A for inapplicable metrics for certain methods. Please refer to App. D for more details.

### 5.1 Synthetic Data

**Data generation.** We evaluate our algorithm on a scalar Merton jump diffusion model. The dataset consists of $N = 10,000$ paths, generated using the standard EM scheme with 100 time steps. Using a sliding window with stride 1, we predict the next 10 frames from the past 10. The data is split into 60% training, 20% validation, and 20% testing. Refer to App. D for details.

**Results.** Tab. 1 reports quantitative results on the jump-driven synthetic dataset. Learning-based methods outperform traditional statistical models (ARIMA, BS, and MJD), and our Neural MJD tops all three evaluation protocols, surpassing Neural BS thanks to its explicit jump modeling objective. Qualitatively, as shown in Fig. 3, our Neural MJD generates larger, realistic jumps, while the baselines produce smoother but less accurate trajectories.

### 5.2 Real-World Business and Financial Data

**Business analytics dataset.** The **SafeGraph&Advan** business analytics dataset combines proprietary data from Advan [87] and SafeGraph [88] to capture daily customer spending at points of interest (POIs) in Texas, USA. It includes time-series features (*e.g.*, visits, spending) and static features (*e.g.*, parking availability) for each POI, along with ego graphs linking each POI to its 10 nearest neighbors. Using a sliding window of 14 input days to predict the next 7, the dataset spans Jan.–Dec. 2023 for training, Jan. 2024 for validation, and Feb.–Apr. 2024 for testing.

**Stock price dataset.** The **S&P 500** dataset [89] is a public dataset containing historical daily prices for 500 major US stocks. It comprises time-series data without additional contextual information. We construct a simple fully connected graph among all listed companies. Similarly to the business analytics dataset, we employ a sliding window approach with a stride of 1, using the past 14 days as input to predict the next 7 days. The dataset is divided into training (Jan.–Dec. 2016), validation (Jan. 2017), and testing (Feb.–Apr. 2017) sets. Refer to App. D for further details about the datasets.

Table 2: Quantitative results on the **SafeGraph&Advan** business analytics dataset.

| Metrics | Mean | | | Winner-takes-all | | | Probabilistic | | |
|---|---|---|---|---|---|---|---|---|---|
| Model | MAE↓ | MSE↓ | $R^2$↑ | minMAE↓ | minMSE↓ | maxR$^2$↑ | $p$-MAE↓ | $p$-MSE↓ | $p$-R$^2$↑ |
| ARIMA | 152.6 | 1.66e05 | -0.183 | | N/A | | | N/A | |
| BS | 135.5 | 1.05e05 | 0.102 | 112.8 | 9.01e04 | 0.159 | 121.5 | 9.87e04 | 0.138 |
| MJD | 131.8 | 9.98e04 | 0.127 | 109.6 | 8.48e04 | 0.169 | 117.6 | 9.02e04 | 0.144 |
| XGBoost | 124.0 | 9.76e04 | 0.303 | | N/A | | | N/A | |
| MLP | 109.5 | 8.18e04 | 0.416 | | N/A | | | N/A | |
| GCN | 95.2 | 7.12e04 | 0.432 | | N/A | | | N/A | |
| DDPM | 68.5 | 4.75e04 | 0.501 | 58.9 | 4.48e04 | 0.529 | | N/A | |
| EDM | 57.6 | 4.35e04 | 0.525 | 49.4 | 3.76e04 | 0.556 | | N/A | |
| FM | 54.5 | 4.32e04 | 0.540 | 47.8 | 3.58e04 | 0.552 | | N/A | |
| NeuralCDE | 94.6 | 7.09e04 | 0.425 | | N/A | | | N/A | |
| LatentSDE | 75.7 | 5.26e04 | 0.487 | 66.5 | 4.58e04 | 0.498 | | N/A | |
| Neural BS | 56.4 | **4.17e04** | 0.539 | 45.6 | 3.45e04 | 0.561 | 55.9 | 4.16e04 | 0.538 |
| Neural MJD (ours) | **54.1** | 4.18e04 | **0.549** | **42.3** | **3.19e04** | **0.565** | **53.0** | **4.10e04** | **0.550** |

Table 3: Quantitative results on the **S&P 500** stock dataset.

| Metrics | Mean | | | Winner-takes-all | | | Probabilistic | | |
|---|---|---|---|---|---|---|---|---|---|
| Model | MAE↓ | MSE↓ | $R^2$↑ | minMAE↓ | minMSE↓ | maxR$^2$↑ | $p$-MAE↓ | $p$-MSE↓ | $p$-R$^2$↑ |
| ARIMA | 62.1 | 3.67e04 | -0.863 | | N/A | | | N/A | |
| BS | 65.1 | 4.01e04 | 0.052 | 44.6 | 1.79e04 | 0.145 | 52.8 | 2.03e04 | 0.105 |
| MJD | 64.3 | 3.58e04 | 0.092 | 40.7 | 1.22e04 | 0.235 | 49.7 | 1.67e04 | 0.112 |
| XGBoost | 44.3 | 1.64e04 | 0.170 | | N/A | | | N/A | |
| MLP | 44.4 | 1.57e04 | 0.205 | | N/A | | | N/A | |
| GCN | 44.7 | 1.53e04 | 0.224 | | N/A | | | N/A | |
| NJ-ODE | 46.8 | 1.69e04 | 0.208 | | N/A | | | N/A | |
| DDPM | 42.2 | 1.88e04 | 0.235 | 36.8 | 8.42e03 | 0.470 | | N/A | |
| EDM | 37.1 | 1.68e04 | 0.249 | 27.6 | 5.01e03 | 0.542 | | N/A | |
| FM | 34.9 | 8.47e03 | 0.368 | 19.8 | 3.59e03 | 0.625 | | N/A | |
| NeuralCDE | 42.8 | 1.46e04 | 0.201 | | N/A | | | N/A | |
| LatentSDE | 39.8 | 1.44e04 | 0.212 | 20.8 | 3.49e03 | 0.617 | | N/A | |
| Neural BS | 31.6 | 4.32e03 | 0.781 | 12.6 | 8.04e02 | 0.959 | 22.3 | 2.19e03 | 0.889 |
| Neural MJD (ours) | **15.4** | **1.36e03** | **0.953** | **4.3** | **1.46e02** | **0.995** | **13.6** | **1.08e03** | **0.963** |

**Results.** Tab. 2 reports results on the SafeGraph&Advan dataset covering POIs revenue prediction, which is measured in dollars. Denoising generative models (*e.g.*, DDPM, EDM, FM) show strong performance, significantly outperforming simple supervised baselines like GCN. Neural MJD further improves upon the strong FM baseline, especially in winner-takes-all metrics, indicating better diversity and accuracy in generating plausible outcomes through simulated jumps. While denoising models support likelihood evaluation, their high computational cost—requiring hundreds of NFEs—makes them unsuitable for large datasets. In contrast, Neural MJD enables fast likelihood evaluation without such overhead, which enables the computation of probabilistic metrics.

Tab. 3 shows similar results on the S&P 500 dataset. FM again outperforms conventional baselines, including ODE based NJ-ODE and NeuralCDE, and Neural MJD achieves the best overall performance, effectively capturing volatility and discontinuities in stock time-series data. For completeness, we also report results for additional deterministic time-series baselines in Appendix D.3.

Table 4: Ablation study on the effect of teacher forcing (TF) and Euler–Maruyama (EM).

| Model | Mean | | Winner-takes-all | | Probabilistic | |
|---|---|---|---|---|---|---|
| | MAE↓ | $R^2$↑ | minMAE↓ | max$R^2$↑ | $p$-MAE↓ | p-$R^2$↑ |
| Ours | 66.7 | 0.495 | 57.4 | 0.511 | 64.5 | 0.499 |
| w. TF | 101.5 | 0.325 | 85.6 | 0.331 | 99.8 | 0.324 |
| w. EM | 85.6 | 0.397 | 79.4 | 0.423 | 84.4 | 0.405 |

Table 5: Runtime comparison.

| Model | Train (ms) | 1-run (ms) | 10-run (ms) |
|---|---|---|---|
| MLP | 65.2 | 52.2 | N/A |
| GCN | 271.3 | 250.7 | N/A |
| FM | 184.6 | 275.4 | 2696.3 |
| Ours | 183.5 | 166.8 | 179.2 |

### 5.3 Ablation Study

We perform ablation studies to evaluate (i) the training algorithm described in Alg. 1 and (ii) the Euler-Maruyama with restart solver introduced in Sec. 4.3. For the ablations, we use 10% of the SafeGraph&Advan business analytics training set for training and evaluate on the full validation set.

The results are presented in Tab. 4. Our training algorithm computes the MLE loss using the model predictions instead of ground truth, unlike teacher forcing. This improves training stability and reduces the generalization gap. Additionally, we empirically show that the vanilla EM solver results in higher variance and worse performance compared to our solver.

Additionally, we compare the runtime of our method against various baselines in Tab. 5. Thanks to the efficient numerical simulation-based forecasting framework that does not increase NFEs, our models are particularly well-suited for efficient multi-run stochastic predictions.

## 6 Conclusion

We introduced *Neural MJD*, a neural non-stationary Merton jump diffusion model for time series forecasting. By integrating a time-inhomogeneous Itô diffusion and a time-inhomogeneous compound Poisson process, our approach captures non-stationary time series with abrupt jumps. We further proposed a likelihood truncation mechanism and an improved solver for efficient training and inference respectively. Experiments demonstrate that *Neural MJD* outperforms state-of-the-art approaches. Future work includes extending to more challenging data types like videos.

## Acknowledgments and Disclosure of Funding

This work was funded, in part, by the NSERC DG Grant (No. RGPIN-2022-04636), the Vector Institute for AI, Canada CIFAR AI Chair, a Google Gift Fund, and the CIFAR Pan-Canadian AI Strategy through a Catalyst award. Resources used in preparing this research were provided, in part, by the Province of Ontario, the Government of Canada through the Digital Research Alliance of Canada `alliance.can.ca`, and companies sponsoring the Vector Institute `www.vectorinstitute.ai/#partners`, and Advanced Research Computing at the University of British Columbia. Additional hardware support was provided by John R. Evans Leaders Fund CFI grant. Y.L. and Y.G. are supported by the NSF grant IIS-2153468. Q.Y. is supported by UBC Four Year Doctoral Fellowship.

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

# Appendix

**TABLE OF CONTENTS**

# A Derivations of the Stationary Merton Jump Diffusion Model

In this section, we briefly review the mathematical derivations from classical textbooks to ensure the paper is self-contained. Our primary focus is on the case where the state variable $S$ is scalar, as is common in many studies. However, in Sec. 4, we extend our analysis to the more general $\mathbb{R}^d$ setting. Notably, in our framework, we do not account for correlations among higher-dimensional variables. For instance, the covariance matrix of the Brownian motion is assumed to be isotropic, meaning all components have the same variance. To maintain clarity and consistency with standard textbook conventions, we adopt scalar notations throughout this section for simplicity.

## A.1 MJD and Lévy Process

**Definition A.1. Lévy process** [65, Definition 3.1] A càdlàg (right-continuous with left limits) stochastic process $(X_t)_{t \geq 0}$ on $(\Omega, \mathcal{F}, \mathbb{P})$ with values in $\mathbb{R}^d$ such that $X_0 = 0$ is called a Lévy process if it possesses the following properties:

1. Independent increments: For every increasing sequence of times $t_0, t_1, \ldots, t_n$, the random variables $X_{t_0}, X_{t_1} - X_{t_0}, \ldots, X_{t_n} - X_{t_{n-1}}$ are independent.

2. Stationary increments: The law of $X_{t+h} - X_t$ does not depend on $t$.

3. Stochastic continuity: For all $\varepsilon > 0$, $\lim_{h \to 0} \mathbb{P}(|X_{t+h} - X_t| \geq \varepsilon) = 0$.

A Lévy process $(X_t)_{t \geq 0}$ is a stochastic process that generalizes jump-diffusion dynamics, incorporating both continuous Brownian motion and discontinuous jumps. The Merton Jump Diffusion (MJD) model given by,

$$\mathrm{d}S_t = S_t((\mu - \lambda k)\mathrm{d}t + \sigma \mathrm{d}W_t + \mathrm{d}Q_t), \tag{15}$$

is a specific example of a Lévy process, as it comprises both a continuous diffusion component and a jump component. According to the Lévy–Itô decomposition [65, Proposition 3.7], any Lévy process can be expressed as the sum of a deterministic drift term, a Brownian motion component, and a pure jump process, which is represented as a stochastic integral with respect to a Poisson random measure.

## A.2 Explicit Solution to MJD

To derive the solution to MJD in Eq. (2), based on [65, Proposition 8.14], we first apply Itô's formula to the SDE:

$$\mathrm{d}f(S_t, t) = \frac{\partial f(S_t, t)}{\partial t}\mathrm{d}t + b_t \frac{\partial f(S_t, t)}{\partial S_t}\mathrm{d}t + \frac{\omega_t^2}{2}\frac{\partial^2 f(S_t, t)}{\partial S_t^2}\mathrm{d}t + \omega_t \frac{\partial f(S_t, t)}{\partial S_t}\mathrm{d}W_t$$
$$+ [f(S_t) - f(S_{t-})], \tag{16}$$

where $b_t = (\mu - \lambda k)S_t$, $\omega_t = \sigma S_t$, and $S_{t-}$ represents the value of $S$ before the jump at time $t$.

By setting the function $f(S_t, t) = \ln S_t$, the formula can be rearranged as:

$$\mathrm{d}\ln S_t = \frac{\partial \ln S_t}{\partial t}\mathrm{d}t + (\mu - \lambda k)S_t \frac{\partial \ln S_t}{\partial S_t}\mathrm{d}t + \frac{\sigma^2 S_t^2}{2}\frac{\partial^2 \ln S_t}{\partial S_t^2}\mathrm{d}t + \sigma S_t \frac{\partial \ln S_t}{\partial S_t}\mathrm{d}W_t$$
$$+ [\ln(S_t) - \ln(S_{t-})]$$
$$= (\mu - \lambda k)S_t \frac{1}{S_t}\mathrm{d}t + \frac{\sigma^2 S_t^2}{2}(\frac{1}{-S_t^2})\mathrm{d}t + \sigma S_t(\frac{1}{S_t})\mathrm{d}W_t + [\ln(S_t) - \ln(S_{t-})]$$
$$= (\mu - \lambda k)\mathrm{d}t - \frac{\sigma^2}{2}\mathrm{d}t + \sigma \mathrm{d}W_t + [\ln(S_t) - \ln(S_{t-})] \tag{17}$$

From the definition of the Compound Poisson process, we have that $S_t = Y_i S_{t-}$, such that $\ln(S_t) - \ln(S_{t-}) = \ln Y_i$. Here, $Y_i$ is the magnitude of the multiplicative jump. Therefore, integrating both sides of Eq. (17), we get the final explicit solution for MJD model:

$$\ln S_t - \ln S_0 = (\mu - \lambda k - \frac{1}{2}\sigma^2)t + \sigma W_t + \sum_{i=1}^{N_t} \ln Y_i. \tag{18}$$

We can reorganize the explicit solution as:

$$S_t = S_0 \exp\left(\left(\mu - \lambda k - \frac{\sigma^2}{2}\right)t + \sigma W_t + \sum_{i=1}^{N_t} \ln Y_i\right), \tag{19}$$

since the drift term, diffusion term and jump term are independent, we can derive the mean of $S_t$ conditional on $S_0$:

$$
\begin{aligned}
\mathbb{E}[S_t|S_0] &= S_0 \mathbb{E}\left[\exp\left(\left(\mu - \lambda k - \frac{\sigma^2}{2}\right)t + \sigma W_t + \sum_{i=1}^{N_t} \ln Y_i\right)\right] \\
&= S_0 \mathbb{E}\left[\exp\left(\left(\mu - \lambda k - \frac{\sigma^2}{2}\right)t\right)\right] \cdot \mathbb{E}\left[\exp\left(\sigma W_t\right)\right] \cdot \mathbb{E}\left[\exp\left(\sum_{i=1}^{N_t} \ln Y_i\right)\right] \\
&= S_0 \exp\left(\left(\mu - \lambda k - \frac{\sigma^2}{2}\right)t\right) \cdot \exp\left(\frac{\sigma^2}{2}t\right) \cdot \exp\left(\lambda k t\right) \\
&= S_0 \exp\left(\mu t\right) \tag{20}
\end{aligned}
$$

### A.3 Likelihood Function of MJD

For the log-likelihood derivation, given the conditional probability in Eq. (4), the log-likelihood of the MJD model can be expressed as:

$$
\begin{aligned}
\log P\left(\ln S_t|S_0\right) &= \log \sum_{n=0}^{\infty} P\left(N_t = n\right) P\left(\ln S_t|S_0, N_t = n\right) \\
&= \log \sum_{n=0}^{\infty} \exp\left(-\lambda t\right) \frac{(\lambda t)^n}{n!} \frac{1}{\sqrt{2\pi b_n^2}} \exp\left(-\frac{(\ln S_t - a_n)^2}{2 b_n^2}\right) \\
&= \log \sum_{n=0}^{\infty} \exp\left(-\lambda t + \log\frac{(\lambda t)^n}{n!} + \log\frac{1}{\sqrt{2\pi b_n^2}} + \left(-\frac{(\ln S_t - a_n)^2}{2 b_n^2}\right)\right) \\
&= \log \sum_{n=0}^{\infty} \exp\left(-\lambda t + n\log\left(\lambda t\right) - \log n! \sqrt{2\pi} - \frac{\log b_n^2}{2} - \frac{(\ln S_t - a_n)^2}{2 b_n^2}\right), \tag{21}
\end{aligned}
$$

where $a_n = \ln S_0 + \left(\mu - \lambda k - \frac{\sigma^2}{2}\right)t + n\nu$ and $b_n^2 = \sigma^2 t + \gamma^2 n$. In maximum likelihood estimation (MLE), the initial asset price $S_0$ is assumed to be constant (non-learnable) and can therefore be excluded from optimization. The objective of MLE is to estimate the parameter set $\Theta = \{\mu, \sigma, \lambda, \gamma, \nu\}$ by maximizing the likelihood of the observed data under the estimated parameters. For the MJD model, the MLE objective is to determine the optimal parameters $\hat{\Theta}$. By omitting constant terms and expanding $s_n$ and $a_n$, the final expression of the MLE objective can be simplified as:

$$
\begin{aligned}
\hat{\Theta} &= \arg\max_{\Theta} \log P(\ln S_t|S_0) \\
&= \arg\max_{\Theta} \log \sum_{n=0}^{\infty} \exp\Big(-\lambda t + n\log\left(\lambda t\right) - \frac{\log\left(\sigma^2 t + n\gamma^2\right)}{2} \\
&\quad - \frac{(\ln S_t - \ln S_0 - \left(\mu - \frac{\sigma^2}{2} - \lambda k\right)t - n\nu)^2}{2\left(\sigma^2 t + n\gamma^2\right)}\Big) \tag{22}
\end{aligned}
$$

According to [90], the Fourier transform can be applied to the Merton Jump Diffusion log-return density function. The characteristic function is then given by:

$$
\begin{aligned}
\phi_c(\omega) &= \int_{-\infty}^{\infty} \exp(i\omega x) P(x) \mathrm{d}x \\
&= \exp\left[\lambda t\left\{\exp\left(i\omega\nu + \frac{\gamma^2\omega^2}{2}\right) - 1\right\} + i\omega\left(\left(\mu - \frac{\sigma^2}{2} - \lambda k\right)t\right) - \frac{\sigma^2\omega^2}{2}t\right], \tag{23}
\end{aligned}
$$

where $x = \ln \frac{S_t}{S_0}$.

With simplification $\phi_c(\omega) = \exp[tg(\omega)]$, we can find the characteristic exponent, namely, the cumulant generating function (CGF):

$$g(\omega) = \lambda \left\{ \exp\left( i\omega\nu + \frac{\gamma^2\omega^2}{2} \right) - 1 \right\} + i\omega \left( \mu - \frac{\sigma^2}{2} - \lambda k \right) - \frac{\sigma^2\omega^2}{2}, \qquad (24)$$

where $k = \exp(\nu + \gamma^2/2) - 1$.

The series expansion of CFG is:

$$g(\omega) = i\omega k_1 - \frac{\omega^2 k_2}{2} + \frac{\omega^3 k_3}{6}... \qquad (25)$$

According to [65, Proposition 3.13], the cumulants of the Lévy distribution increase linearly with $t$. Therefore, the first cumulant $k_1$ is the mean of the standard MJD:

$$k_1(t) = \mathbb{E}\left[ \ln \frac{S_t}{S_0} \right] = (\mu - \lambda k - \sigma^2/2 + \lambda\nu)t \qquad (26)$$

The second cumulant $k_2$ is variance of the standard MJD, which is:

$$k_2(t) = \mathrm{Var}\left[ \ln \frac{S_t}{S_0} \right] = \left( \sigma^2 + \lambda(\gamma^2 + \nu^2) \right) t \qquad (27)$$

The corresponding higher moments can also be calculated as:

$$\text{Skewness} = k_3(t) = \lambda(3\gamma^2\nu + \nu^3)t \qquad (28)$$

$$\text{Excess Kurtosis} = k_4(t) = \lambda(3\gamma^4 + 6\nu^2\gamma^2 + \nu^4)t \qquad (29)$$

# B   Derivations of the Non-stationary Merton Jump Diffusion Model

## B.1   Non-stationary MJD and Additive Process

**Definition B.1. Additive process** [65, Definition 14.1] A stochastic process $(X_t)_{t \geq 0}$ on $\mathbb{R}^d$ is called an additive process if it is càdlàg, satisfies $X_0 = 0$, and has the following properties:

1. Independent increments: For every increasing sequence of times $t_0, t_1, \ldots, t_n$, the random variables $X_{t_0}, X_{t_1} - X_{t_0}, \ldots, X_{t_n} - X_{t_{n-1}}$ are independent.

2. Stochastic continuity: For all $\varepsilon > 0$, $\lim_{h \to 0} \mathbb{P}(|X_{t+h} - X_t| \geq \varepsilon) = 0$.

In the non-stationary MJD model, given by,

$$dS_t = S_t((\mu_t - \lambda_t k_t)dt + \sigma_t dW_t + \int_{\mathbb{R}^d} (y - 1)N(dt, dy)), \qquad (30)$$

the parameters governing drift, volatility, and jump intensity evolve over time, resulting in non-stationary increments. This violates the key stationarity property required for Lévy processes, as discussed in App. A. Consequently, the non-stationary MJD no longer falls within the Lévy process framework. Instead, according to the definition above, a stochastic process with independent increments that follow a non-stationary distribution is classified as an additive process. Similar to the relationship between the stationary MJD and the Lévy process, the non-stationary MJD can be viewed as a specific instance of an additive process. Thus, we can apply corresponding mathematical tools for additive processes to study the non-stationary MJD.

## B.2   Explicit Solution to Non-stationary MJD

To derive the explicit solution to the non-stationary MJD, according to [65, Proposition 8.19], we have the Itô formula for semi-martingales:

$$f(t, X_t) - f(0, X_0) = \int_0^t \frac{\partial f(s, X_s)}{\partial s}ds + \int_0^t \frac{\partial f(s, X_{s-})}{\partial x}dX_s + \frac{1}{2}\int_0^t \frac{\partial^2 f(s, X_{s-})}{\partial x^2}d[X, X]_s^c$$

$$+ \sum_{0 \leq s \leq t, \triangle X_s \neq 0} [f(s, X_s) - f(s, X_{s-}) - \triangle X_s \frac{\partial f(s, X_{s-})}{\partial x}]. \qquad (31)$$

According to [65, Remark 8.3], for a function independent of time (*i.e.*, $f(t, X_t) = f(X_t)$), when we have finite number of jumps, we can rewrite the above equation as:

$$f(X_t) - f(X_0) = \int_0^t f'(X_{s-})\mathrm{d}X_s^c + \frac{1}{2}\int_0^t f''(X_{s-})\mathrm{d}[X,X]_s^c + \sum_{0 \le s \le t, \triangle X_s \ne 0}[f(X_s) - f(X_{s-})],$$

where $X_s^c$ is the continuous part of $X_s$, and $[X,X]_s^c$ is the continuous quadratic variation of $X$ over the interval $[0, s]$.

In our case, let $X_t = S_t$, and define $f(t, X_t) = \ln S_t$, the corresponding derivatives are $\frac{\partial f(t, X_t)}{\partial X_t} = \frac{\partial \ln S_t}{\partial S_t} = \frac{1}{S_t}$, and $\frac{\partial^2 f(t, X_t)}{\partial X_t^2} = \frac{1}{-S_t^2}$. The dynamics of non-stationary MJD is defined by:

$$\mathrm{d}S_t = S_t\left(\mu_t\mathrm{d}t - \lambda_t k_t\mathrm{d}t + \sigma_t\mathrm{d}W_t + \int_{\mathbb{R}^d}(y-1)N(\mathrm{d}t, \mathrm{d}y)\right).$$

The continuous part of the quadratic variation of $S_t$ is $\mathrm{d}[S,S]_s^c = S_t^2\sigma_t^2\mathrm{d}t$. A jump at time $s$ is modeled as a multiplicative change $S_s = yS_{s-}$. Thus, the jump contribution is $\sum_{0 \le s \le t, \triangle X_s \ne 0}[f(X_s) - f(X_{s-})] = \sum_{0 \le s \le t, \triangle X_s \ne 0}[\ln(yS_{s-}) - \ln(S_{s-})] = \sum_{0 \le s \le t, \triangle X_s \ne 0}[\ln y]$. Since the jump process is driven by a Poisson random measure $N(\mathrm{d}t, \mathrm{d}y)$ on $[0, t] \times \mathbb{R}^d$, we can rewrite the sum over all jump times as an integral with respect to this measure. When there are finitely many jumps on $[0, t]$, we have $\sum_{0 \le s \le t, \triangle X_s \ne 0}[\ln y] = \int_0^t\int_{\mathbb{R}^d}\ln yN(\mathrm{d}s, \mathrm{d}y)$.

Based on [65, Ch. 14], even when the parameters (drift, volatility, jump intensity, etc.) are time-dependent, the non-stationary MJD remains a semi-martingale. Therefore, we can simplify the equation as follows for time $t$:

$$\ln S_t - \ln S_0 = \int_0^t\frac{1}{S_s}S_s(\mu_s - \lambda_s k_s)\,\mathrm{d}s - \frac{1}{2}\int_0^s\frac{1}{S_s^2}S_s^2\sigma_s^2\mathrm{d}s + \int_0^t\sigma_s\mathrm{d}W_s$$
$$+ \int_{[0,t]\times\mathbb{R}^d}\ln yN(\mathrm{d}s, \mathrm{d}y) \tag{32}$$

Therefore, the explicit solution is:

$$\ln\frac{S_t}{S_0} = \int_0^t(\mu_s - \lambda_s k_s - \frac{\sigma_s^2}{2})\mathrm{d}s + \int_0^t\sigma_s\mathrm{d}W_s + \int_0^t\int_{\mathbb{R}^d}\ln yN(\mathrm{d}s, \mathrm{d}y). \tag{33}$$

The only assumption needed for the derivation is the finite variation condition: $\int_0^t\int_{\mathbb{R}}|y|N(\mathrm{d}s, \mathrm{d}y) < \infty$. Based on the explicit solution for $S_t$, we can easily compute the conditional expectations as,

$$\mathbb{E}[\ln(S_t/S_0)] = \int_0^t(\mu_s - \lambda_s k_s - \frac{\sigma_s^2}{2})\mathrm{d}s + \int_0^t\int_{\mathbb{R}^d}\ln y\lambda_s f_Y(s, y)\mathrm{d}y\mathrm{d}s. \tag{34}$$

and

$$\mathbb{E}[S_t|S_0] = S_0\exp(\int_0^t\mu_s\mathrm{d}s), \tag{35}$$

The variance can also be calculated as,

$$\mathrm{Var}[\ln(S_t/S_0)] = \int_0^t\sigma_s^2\mathrm{d}s + \int_0^t\int_{\mathbb{R}^d}(\ln y)^2\lambda_s f_Y(s, y)\mathrm{d}y\mathrm{d}s. \tag{36}$$

Given the results for the general time-inhomogeneous system, one can directly substitute the coefficients into the discrete formulation implemented in Sec. 4.2 to obtain the corresponding results.

## B.3 Likelihood Function of Non-stationary MJD

Let $X_t = \ln S_t/\ln S_0, t \ge 0$ be the log-return of the asset price $S_t$. Under the non-stationary MJD settings, $X_t$ is an additive process, therefore by the general property of additive process [65, Ch 14], the law of $X_t$ is infinitely divisible and its characteristic function is given by the Lévy–Khintchine formula:

$$\mathbb{E}[\exp(iu \cdot X_t)] = \exp\psi_t(u),$$

where

$$\psi_t(u) = -\frac{1}{2}u \cdot A_t u + iu \cdot \Gamma_t + \int_{\mathbb{R}^d} \eta(dy)\left(e^{iu\cdot x} - 1 - iu \cdot x\right), \tag{37}$$

where we have the integrated volatility term $A_t = \int_0^t \sigma_s ds$, the integrated drift term $\Gamma_t = \int_0^t (\mu_s - \lambda_s k_s)ds$, and the Lévy measure $\eta(dy) = \lambda_t f_Y(t, y)$.

Since the jumps follow a time-inhomogeneous Poisson random measure and the process is additive, we can denote the integrated intensity of jumps by $\Lambda(t) = \int_0^t \lambda_s ds$, then the number of jumps $N_t$ in the time range $[0, t]$ is a Poisson distribution with this integrated jump intensity $\Lambda(t)$. When conditioning on $N_t$, we will have:

$$P(N_t = n) = \frac{\exp(-\Lambda(t))\Lambda(t)^n}{n!} \tag{38}$$

We now derive the conditional density $P(\ln S_t | N_t = n, S_0)$, and here we can start with the case of one jump. When there is exactly one jump in $[0, t]$, the jump time $s_1$ is random. Given jump time $s_1$, in a time-inhomogeneous setting, the instantaneous probability of a jump at time $s_1$ is proportional to $\lambda_{s_1}$. According to the dynamics of non-stationary MJD, the continuous part of the log-return leads to a normal distribution with mean being $a_1 = \ln S_0 + \int_0^{s_1}\left(\mu_s - \lambda_s k_s - \frac{\sigma_s^2}{2}\right)ds + \nu_{s_1}$, and variance being $b_1^2 = \int_0^{s_1} \sigma_s^2 ds + \gamma_{s_1}^2$. Thus, the conditional density of $\ln S_t$ given one jump at time $s_1$ is $\phi(\ln S_t; a_1, b_1^2)$, where $\phi(\cdot; a_1, b_1^2)$ denotes the Gaussian density with mean $a_1$ and variance $b_1^2$.

Since the jump could have occurred at any time in $[0, t]$, we must integrate over the possible jump time $s_1$. Therefore, the conditional density given $N_t = 1$ is:

$$P(\ln S_t | N_t = 1, S_0) = \frac{1}{\Lambda(t)}\int_0^t \lambda_{s_1}\phi(\ln S_t; a_1, b_1^2)\, ds_1, \tag{39}$$

where $\frac{1}{\Lambda(t)}$ normalizes the density.

When generalizing to the case of $N^t = n$, the conditional density $P(\ln S_t | N_t = n, S_0)$ is defined via an integration over the $n$ jump times, with the jump times denoted by $0 \le s_1, ..., s_n \le t$.

Because the process is time-inhomogeneous, the probability density that a jump occurs at a specific time $s_i$ is given by the instantaneous rate $\lambda_{s_i}$, therefore for a given set of jump times, the joint density for the jumps is proportional to $\Pi_{i=1}^n \lambda_{s_i}$. The conditional density can be written as:

$$P(\ln S_t | N_t = n, S_0) = \frac{1}{\Lambda(t)^n}\int \cdots \int_{[0,t]} \Pi_{i=1}^n \lambda_{s_i}\phi(\ln S_t; a_n, b_n^2)\, ds_1 \cdots ds_n \tag{40}$$

Here $\phi(\ln S_t; a_n, b_n^2)$ is the density of a normal distribution with mean $a_n$ and variance $b_n^2$, which are defined by:

$$a_n = \ln S_0 + \int_0^t\left(\mu_s - \lambda_s k_s - \frac{\sigma_s^2}{2}\right)ds + \sum_{i=1}^n \nu_{s_i}$$

$$b_n^2 = \int_0^t \sigma_s^2 dt + \sum_{i=1}^n \gamma_{s_i}^2$$

For convenience, we may write the mixture term as

$$\Phi_n = \int \cdots \int_{[0,t]} \Pi_{i=1}^n \lambda_{s_i}\phi(\ln S_t; a_n, b_n^2)\, ds_1 \cdots ds_n \tag{41}$$

Therefore, for the time-varying SDEs, the conditional probability of $\ln S_t$ is given by,

$$P(\ln S_t | S_0) = \sum_{n=0}^{\infty} \frac{\exp(-\Lambda(t))}{n!}\Phi_n. \tag{42}$$

# C  Proofs of Theorem and Proposition

## C.1  Proof of Theorem 4.1

**Theorem 4.1.** *Let the likelihood approximation error in Eq.* (12)*, truncated to at most $\kappa$ jumps, be*

$$\Psi_\kappa(t, \delta) \coloneqq \sum_{n=\kappa+1}^\infty P\left(\Delta N = n\right) P\left(\ln S_{t+\delta} \mid S_t, \mathcal{C}, \Delta N = n\right).$$

*Then, $\Psi_\kappa(t, \delta)$ decays at least super-exponentially as $\kappa \to \infty$, with a convergence rate of $O(\kappa^{-\kappa})$.*

Before diving into the proof, we first introduce two important lemmas.

**Lemma C.1** (Theorem 2 in [2])**.** *Let $Y \sim \mathrm{Pois}(m)$ be a Poisson-distributed random variable with mean $m$. Its distribution function is defined as $P(Y \leq k) \coloneqq \exp(-m) \sum_{i=0}^k \frac{m^i}{i!}$, with integer support $k \in \{0, 1, \ldots, \infty\}$. For $k = 0$ and $k = \infty$, one has $P(Y \leq 0) = \exp(-m), P(Y \leq \infty) = 1$. For every other $k \in \{1, 2, 3, \ldots\}$, the following inequalities hold:*

$$\Phi\left(\mathrm{sign}(k - m)\sqrt{2H(m, k)}\right) < P(Y \leq k) < \Phi\left(\mathrm{sign}(k + 1 - m)\sqrt{2H(m, k + 1)}\right),$$

*where $H(m, k)$ is the Kullback-Leibler (KL) divergence between two Poisson-distributed random variables with respective means $m$ and $k$:*

$$H(m, k) = D_{KL}\left(\mathrm{Pois}(m)\|\mathrm{Pois}(k)\right) = m - k + k\ln\left(\frac{k}{m}\right).$$

*And $\Phi(x)$ is the cumulative distribution function (CDF) of the standard normal distribution and $\mathrm{sign}(\cdot)$ is the signum function.*

Lemma C.1 is particularly helpful in our proof below. We also acknowledge its foundation in an earlier work [91], which provides many insights and a profound amount of valuable knowledge on its own.

**Lemma C.2** (Bounds on the Standard Normal CDF)**.** *The following upper bound for $\Phi(\cdot)$ holds when $x < 0$:*

$$\Phi(x) < \frac{\phi(x)}{|x|},$$

*where $\phi(x) = \frac{\exp(-x^2/2)}{\sqrt{2\pi}}$ is the probability density function of the standard normal distribution.*

*Proof.* By the Mills' ratio inequality for the Gaussian distribution [92], we have $1 - \Phi(x) < \frac{\phi(x)}{x}, \forall x > 0$. Using the identity $\Phi(-x) = 1 - \Phi(x)$ for $x > 0$, we immediately obtain: $\Phi(-x) < \frac{\phi(x)}{x}, \forall x > 0$. For $x < 0$, substituting $-x$ into the previous bound and noting that $\phi(-x) = \phi(x)$, we obtain $\Phi(x) < \frac{\phi(x)}{|x|}, \forall x < 0$.  $\square$

**Proof of Theorem 4.1.**

*Proof.* The original likelihood objective in Eq. (12) is as follows:

$$
\begin{aligned}
P\left(\ln S_{t+\delta}|S_t, \mathcal{C}\right) &= \sum_{n=0}^\infty P\left(\Delta N = n\right) P\left(\ln S_{t+\delta}|S_t, \mathcal{C}, \Delta N = n\right) \\
&= \sum_{n=0}^\infty \exp\left(-\lambda_{\rho_t}\delta\right) \frac{(\lambda_{\rho_t}\delta)^n}{n!} \phi\left(\ln S_{t+\delta}; a_{n,\delta}, b_{n,\delta}^2\right) \\
&= \sum_{n=0}^\infty \exp\left(-\lambda_{\rho_t}\delta\right) \frac{(\lambda_{\rho_t}\delta)^n}{n!} \frac{1}{\sqrt{2\pi b_{n,\delta}^2}} \exp\left(-\frac{(\ln S_{t+\delta} - a_{n,\delta})^2}{2b_{n,\delta}^2}\right)
\end{aligned}
\tag{43}
$$

where $\delta$ is a small time change so that $\rho_t - 1 \leq t < t + \delta < \rho_t$, $a_{n,\delta} = \ln S_t + (\mu_{\rho_t} - \lambda_{\rho_t}k_{\rho_t} - \sigma_{\rho_t}^2/2)\delta + n\nu_{\rho_t}$ and $s_{n,\delta}^2 = \sigma_{\rho_t}^2\delta + \gamma_{\rho_t}^2 n$.

We define the truncation error with a threshold $\kappa$ as:

$$\Psi_\kappa(t,\delta) := \sum_{n=\kappa+1}^{\infty} P\left(\Delta N = n\right) P\left(\ln S_{t+\delta} \mid S_t, \mathcal{C}, \Delta N = n\right). \tag{44}$$

The second term $P\left(\Delta N = n\right) P\left(\ln S_{t+\delta} \mid S_t, \mathcal{C}, \Delta N = n\right)$ is a Gaussian density function and upper bounded by $\frac{1}{\sqrt{2\pi b_{n,\delta}^2}}$, so the truncation error $\Psi_\kappa(t,\delta)$ is bounded by:

$$\Psi_\kappa(t,\delta) \leq \sum_{n=\kappa+1}^{\infty} \frac{1}{\sqrt{2\pi b_{n,\delta}^2}} P\left(\Delta N = n\right) \qquad \text{(Gaussian density bound)}$$

$$\leq \frac{1}{\sqrt{2\pi b_{\kappa+1,\delta}^2}} \sum_{n=\kappa+1}^{\infty} P\left(\Delta N = n\right) \qquad (b_{\kappa,\delta} \text{ increases as } \kappa \text{ goes up})$$

$$= \frac{1}{\sqrt{2\pi b_{\kappa+1,\delta}^2}} \left(1 - \sum_{n=0}^{\kappa} P\left(\Delta N = n\right)\right) \qquad \text{(property of Poisson CDF)}$$

$$< \frac{1}{\sqrt{2\pi b_{\kappa+1,\delta}^2}} \left(1 - \Phi(\text{sign}(\kappa - \lambda_{\rho_t}\delta)\sqrt{2D_{\text{KL}}(\text{Pois}(\lambda_{\rho_t}\delta)\|\text{Pois}(\kappa))})\right) \qquad \text{(Lemma C.1)}$$

$$= \frac{1}{\sqrt{2\pi b_{\kappa+1,\delta}^2}} \Phi(\text{sign}(\lambda_t\delta - \kappa)\sqrt{2D_{\text{KL}}(\text{Pois}(\lambda_{\rho_t}\delta)\|\text{Pois}(\kappa))}) \qquad \text{(Gaussian CDF)}$$

As stated above, the KL divergence between two Poisson distributions follows

$$D_{\text{KL}}\left(\text{Pois}(a)\|\text{Pois}(b)\right) = a - b + b\ln(\frac{b}{a})$$

Therefore,

$$\Psi_\kappa(t,\delta) < \frac{1}{\sqrt{2\pi b_{\kappa+1,\delta}^2}} \Phi\left(\text{sign}(\lambda_{\rho_t}\delta - \kappa)\sqrt{2D_{\text{KL}}(\text{Pois}(\lambda_{\rho_t}\delta)\|\text{Pois}(\kappa))}\right),$$

$$= \frac{1}{\sqrt{2\pi(\sigma_{\rho_t}^2\delta + \gamma_{\rho_t}^2(\kappa+1))}} \Phi\left(\text{sign}(\lambda_{\rho_t}\delta - \kappa)\sqrt{2(\lambda_{\rho_t}\delta - \kappa + \kappa\ln(\frac{\kappa}{\lambda_{\rho_t}\delta}))}\right)$$

$$= \frac{1}{\sqrt{2\pi(\sigma_{\rho_t}^2\delta + \gamma_{\rho_t}^2(\kappa+1))}} \Phi\left(\text{sign}(\frac{\lambda_{\rho_t}\delta}{\kappa} - 1)\sqrt{2(\lambda_{\rho_t}\delta - \kappa - \kappa\ln(\frac{\lambda_{\rho_t}\delta}{\kappa}))}\right)$$

Intuitively, the truncation error decreases to zero as $\kappa$ approaches infinity. Below, we analyze the convergence rate. When $\kappa$ is sufficiently large, the term $\text{sign}\left(\frac{\lambda_{\rho_t}\delta}{\kappa} - 1\right)$ is negative. Consequently, the upper bound becomes:

$$\Psi_\kappa(t,\delta) < \frac{1}{\sqrt{2\pi(\sigma_{\rho_t}^2\delta + \gamma_{\rho_t}^2(\kappa+1))}} \Phi\left(-\sqrt{2(\lambda_{\rho_t}\delta - \kappa - \kappa\ln(\frac{\lambda_{\rho_t}\delta}{\kappa}))}\right)$$

$$< \frac{1}{\sqrt{2\pi(\sigma_{\rho_t}^2\delta + \gamma_{\rho_t}^2(\kappa+1))}} \frac{\phi\left(-\sqrt{2(\lambda_{\rho_t}\delta - \kappa - \kappa\ln(\frac{\lambda_{\rho_t}\delta}{\kappa}))}\right)}{\sqrt{2(\lambda_{\rho_t}\delta - \kappa - \kappa\ln(\frac{\lambda_{\rho_t}\delta}{\kappa}))}} \qquad \text{(Lemma C.2)}$$

$$= \frac{\exp\left(-\lambda_{\rho_t}\delta + \kappa + \kappa\ln(\frac{\lambda_{\rho_t}\delta}{\kappa})\right)}{2\pi\sqrt{2(\sigma_{\rho_t}^2\delta + \gamma_{\rho_t}^2(\kappa+1))(\lambda_{\rho_t}\delta - \kappa - \kappa\ln(\frac{\lambda_{\rho_t}\delta}{\kappa}))}}$$

As $\kappa \to \infty$, the numerator is dominated by $\exp(-\kappa\ln\kappa)$, which decays super-exponentially (faster than any polynomial or exponential decay). The denominator consists of two components:

- The first term, $\sqrt{\sigma_{\rho_t}^2 \delta + \gamma_{\rho_t}^2 (\kappa + 1)}$, scales asymptotically as $\gamma_{\rho_t} \sqrt{\kappa}$.

- The second term, $\sqrt{\left( \lambda_{\rho_t} \delta - \kappa - \kappa \ln \left( \frac{\lambda_{\rho_t} \delta}{\kappa} \right) \right)}$, scales as $\sqrt{\kappa \ln \kappa}$.

Combining all terms, the upper bound scales as:

$$\frac{1}{2\pi \sqrt{2} \gamma_{\rho_t}} \cdot \frac{\exp(-\kappa \ln \kappa)}{\kappa \sqrt{\ln \kappa}} \sim \frac{\kappa^{-\kappa}}{\kappa \sqrt{\ln \kappa}}.$$

The term $\kappa^{-\kappa}$ decays super-exponentially, while the denominator grows algebraically (as $\kappa \sqrt{\ln \kappa}$). The rapid decay of $\kappa^{-\kappa}$ dominates the polynomial growth in the denominator. The overall convergence rate is super-exponentially fast, at the rate of $O(\exp(-\kappa \ln \kappa))$ or equivalently $O(\kappa^{-\kappa})$.

Since the upper bound of $\Psi_\kappa(t, \delta)$ decays at the rate of $O(\kappa^{-\kappa})$ and $\Psi_\kappa(t, \delta)$ is strictly positive, this implies that the original quantity $\Psi_\kappa(t, \delta)$ must decay at least as fast as the upper bound. This completes the proof.

$\square$

## C.2 Proof of Proposition 4.2

**Proposition C.3.** *Let $1/M$ be the step size. Both standard EM and our solver exhibit a weak convergence rate of $O(1/M)$. Specifically, the vanilla EM has a weak error of $\epsilon_t^E \leq K \exp(Lt)/M$ for some constant $L > 0$, while ours achieves a tighter weak error of $\epsilon_t^R \leq K \exp(L(t - \lfloor t \rfloor))/M$.*

*Proof.* Here, we prove that this restart strategy has a tighter weak-convergence error than the standard EM solver. Recall that we let $\epsilon_t := |\mathbb{E}[g(\bar{S}_t)] - \mathbb{E}[g(S_t)]|$ be the standard weak convergence error [75], where $S_t$ is the ground truth state, $\bar{S}_t$ is the estimated one using certain sampling scheme and $g$ is a $K$-Lipschitz continuous function. We denote the weak convergence errors of our restarted solver and the standard EM solver by $\epsilon_t^R$ and $\epsilon_t^E$, respectively.

**Step 1: Standard EM Results on Time-Homogeneous MJD SDEs**

Early works on jump-diffusion SDE simulations explored the weak error bounds, which we summarize as follows. For time-homogeneous MJD SDEs, the error term $\epsilon_t^E$ of the standard EM method is dominated by $\epsilon_t^E \leq K \exp(Lt)/M$. This is supported by the following: (a) Theorem 2.2 in [76] establishes the $O(1/M)$ rate; (b) Sec. 4-5 of [76] and Theorem 2.1 of [93] shows that $\epsilon_t^E$ grows exponentially regarding time with a big-$O$ factor $O(e^{K_p(t)})$. In particular, the time-dependent term in the error bound $e^{K_p(t)}$ used in the proof of [76] is rooted in their Lemma 4.1, which can be proven in a more general setting in [93]; *e.g.*, Eq. (2.16) in [93] discusses concrete forms of $K_p(t)$, which can be absorbed into $O(e^{Lt})$ for some constant $L > 0$. Lastly, the $K$-Lipschitz condition of the function $g$ provides the coefficient $K$ in the bound. For a detailed proof—which is more involved and not central to the design and uniqueness of our algorithm—we refer the reader to [75, 94]. When combining the above existing results from the literature, we can derive the error bound of $\epsilon_t^E \leq K \exp(Lt)/M$.

**Step 2: Standard EM Results on Time-Inhomogeneous MJD SDEs**

Our paper considers time-inhomogeneous MJD SDEs, with parameters fixed within each interval $[\tau - 1, \tau)$ ($\tau \in \mathbb{N}$, $\tau \geq 1$). This happens to align with the Euler-Peano scheme for general time-inhomogeneous SDEs approximation. As a specific case of time-varying Lévy processes, our MJD SDEs retain the same big-$O$ bounds as the time-homogeneous case. Namely, the standard EM solver has the same weak convergence error $\epsilon_t^E \leq K \exp(Lt)/M$, as in the time-homogeneous MJD SDEs. This can be justified by extending Section 5 of [76] that originally proves the EM's weak convergence for time-homogeneous Lévy processes. Specifically, the core technique lies in the Lemma 4.1 of [76], which, based on [93], is applicable to both time-homogeneous and Euler-Peano-style inhomogeneous settings (see Remark 3.3.3 in [93]). Therefore, equivalent weak convergence bounds could be attained by extending Lemma 4.1 of [76] with proofs from [93] thanks to the Euler-Peano formulation.

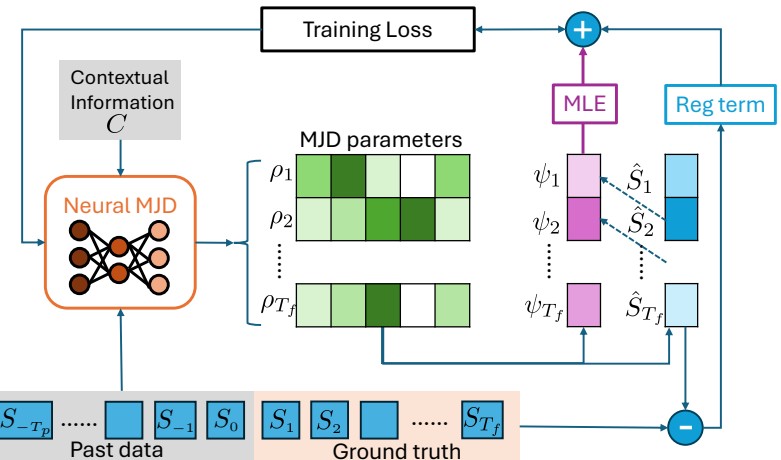

Figure 4: Neural MJD training pipeline. The symbol $\rho$ represents the MJD parameters $\{\mu_\tau, \sigma_\tau, \lambda_\tau, \nu_\tau, \gamma_\tau\}$ in our model.

**Step 3: Our Restarted EM Solver Error Bound**

We now discuss the error bound for the restarted EM solver, $\epsilon_t^R$. Thanks to explicit solutions for future states $\{S_1, S_2, \ldots, S_{T_f}\}$, we can analytically compute their mean $\mathbb{E}[S_\tau|\mathcal{C}]$, $\tau \geq 1$, based on Eq. (13), which greatly simplifies the analysis. Using the restart mechanism in line 10 of Alg. 2, we ensure that $\mathbb{E}[\bar{S}_\tau|\mathcal{C}]$ from our restarted EM solver closely approximates the true $\mathbb{E}[S_\tau|\mathcal{C}]$ at restarting times. $\epsilon_t^R$ is significantly reduced when restart happens (when $t$ is an integer in our context for simplicity), then it grows again at the same rate as the standard EM method until the next restart timestep. This explains the $O(e^{t-\lfloor t \rfloor})$ difference in the error bounds of $\epsilon_t^R$ and $\epsilon_t^E$, where $\lfloor t \rfloor$ is the last restart time. Note that we could make the restart timing more flexible to potentially achieve a tighter bound in terms of weak convergence. However, this may affect the diversity of the simulation results, as the fidelity of path stochasticity could be impacted. □

## D  Experiment Details

### D.1  Baseline, Model Architecture, and Experiment Settings

For the statistical BS and MJD baselines, we assume a stationary process and estimate the parameters using a numerical MLE objective based on past sequences. For the other deep learning baselines, including DDPM, EDM, FM, Neural BS, and Neural MJD, we implement our network using the standard Transformer architecture [25]. All baseline methods are based on the open-source code released by their authors, with minor modifications to adapt to our datasets. Note that the technical term *diffusion* in the context of SDE modeling (*e.g.*, Merton jump diffusion) should not be conflated with diffusion-based generative models [45]. While both involve SDE-based representations of data, their problem formulations and learning objectives differ significantly.

We illustrate the training loss computation pipeline for Neural MJD in Fig. 4. Notably, the loss computation can be processed in parallel across the future time-step horizon, eliminating the need for recursive steps during training. We normalize the raw data into the range of $[0, 1]$ for stability and use a regularization weight $\omega = 1.0$ during training. All experiments were run on NVIDIA A40 and A100 GPUs (48 GB and 80 GB VRAM, respectively).

### D.2  Datasets Details

For all datasets, we normalize the input data using statistics computed from the training set. For non-denoising models, normalization maps the data to the range $[0, 1]$. In contrast, for denoising models (DDPM, EDM, FM), we scale the data to $[-1, 1]$ to align with standard settings used in image generation. Importantly, normalization coefficients are derived solely from the training set statistics. Further details on this process are provided below.

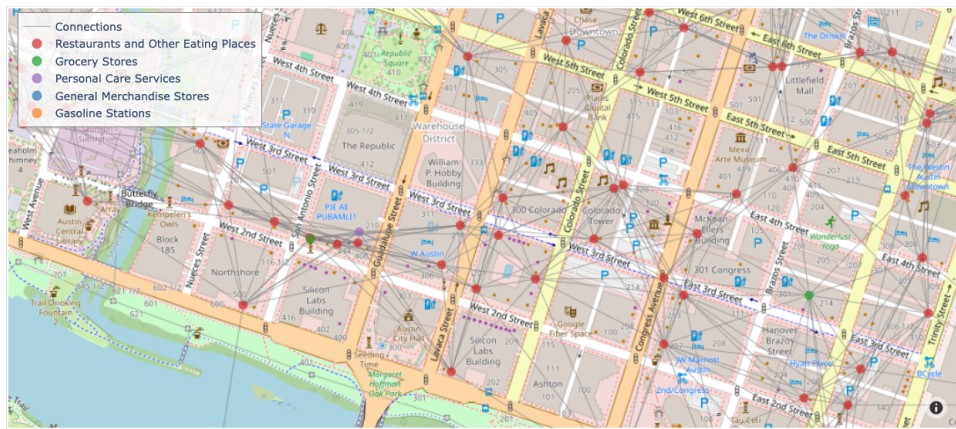

Figure 5: Visualization of Ego Graph Dataset Construction in Austin, Texas

**Synthetic Data.** We generate synthetic data using a scalar Merton Jump Diffusion model. The dataset consists of $N = 10,000$ paths over the interval $[0, 1]$, simulated using the Euler scheme with 100 time steps. To facilitate time-series forecasting, we employ a sliding window approach with a stride of 1, where the model predicts the next 10 frames based on the previous 10. The dataset is divided into 60% training, 20% validation, and 20% testing. For each simulation, model parameters are randomly sampled from uniform distributions: $\mu \sim U(0.1, 0.5)$, $\sigma \sim U(0.1, 0.5)$, $\lambda \sim U(3, 10)$, $\nu \sim U(-0.1, 0.1)$, and $\gamma \sim U(0.5, 1.0)$. These parameter choices ensure the presence of jumps, capturing the stochastic nature of the process.

**SafeGraph&Advan Business Analytics Data.** The SafeGraph&Advan business analytics dataset is a proprietary dataset created by integrating data from Advan [87] and SafeGraph [88] to forecast daily customer spending at points of interest (POIs) across Texas, USA. Both datasets are licensed through Dewey Data Partners under their proprietary commercial terms, and we comply fully with the terms. For each POI, the dataset includes time-series data with dynamic features and static attributes. Additionally, ego graphs are constructed based on geodesic distances, where each POI serves as a central node connected to its 10 nearest neighbors. An visualization is shown in Fig. 5. Specifically, we use POI area, brand name, city name, top and subcategories (based on commercial behavior), and parking lot availability as static features. The dynamic features include spending data, visiting data, weekday, opening hours, and closing hours. These features are constructed for both ego and neighboring nodes. Based on the top category, we determine the maximum spending in the training data and use it to normalize the input data for both training and evaluation, ensuring a regularized numerical range. For training stability, we clip the minimum spending value to 0.01 instead of 0 to enhance numerical stability for certain methods.

We adopt a sliding window approach with a stride of 1, using the past 14 days as input to predict spending for the next 7 days. The dataset spans multiple time periods: the training set covers January–December 2023, the validation set corresponds to January 2024, and the test set includes February–April 2024. This large-scale dataset consists of approximately 3.9 million sequences for training, 0.33 million for validation, and 0.96 million for testing.

**S&P 500 Stock Price Data.** The S&P 500 dataset [89] is a publicly available dataset from Kaggle that provides historical daily stock prices for 500 of the largest publicly traded companies in the U.S (CC0 1.0 Universal license). It primarily consists of time-series data with date information and lacks additional contextual attributes. We include all listed companies and construct a simple fully connected graph among them. Therefore, for models capable of handling graph data, such as GCN, our implemented denoising models, and Neural MJD, we make predictions for all companies (represented as nodes) simultaneously. This differs from the ego-graph processing used in the SafeGraph&Advan dataset, where predictions are made only for the central node, while neighbor nodes serve purely as contextual information. To normalize the data, we determine the maximum stock price for each company in the training data, ensuring that input values fall within the $[0, 1]$ range during training.

Following the approach used for the business analytics dataset, we apply a sliding window method with a stride of 1, using the past 14 days as input to predict stock prices for the next 7 days. The dataset is split into training (Jan.–Dec. 2016), validation (Jan. 2017), and testing (Feb.–Apr. 2017)

sets. In total, it contains approximately 62K sequences for training, 5K for validation, and 15K for testing. To better distinguish the effects of different methods on the S&P 500 dataset, we use a adjusted $R^2$ score $R^2 = 1 - (1 - R^2_{\text{reg}}) \cdot \frac{n-1}{n-p-1}$, where $n$ is the sample size and we set the number of explanatory variables $p$ to be $(k-1)(n-1)/k$, where $k = 70.0$.

### D.3 Additional Deterministic Time-Series Baselines (Third-Party Implementations)

For completeness, we also report results from third-party implementations of Autoformer [95], TiDE [96], and N-HiTS [97], provided by the `NeuralForecast` library (NIXTLA). Results in the table were produced with the publicly available `NeuralForecast` package on the same train/validation/test splits and identical data input (*e.g.* exogenous stock ticker information) as our main experiments, using the package's default training settings without modification.

Table 6: Quantitative results from `NeuralForecast` (NIXTLA) implementations on the **S&P 500** stock dataset.

| Model | MAE $\downarrow$ | MSE $\downarrow$ | $R^2 \uparrow$ |
|---|---|---|---|
| Autoformer | 81.0 | 2.73e04 | 0.061 |
| TiDE | 27.7 | 7.28e03 | 0.750 |
| N-HiTS | 15.5 | 1.86e03 | 0.936 |

### D.4 Limitations

Our approach explicitly models discontinuities (jumps) in the time series. Consequently, if the underlying data lack such jump behaviors—i.e., if they are extremely smooth and exhibit no abrupt shifts—our jump component may be inaccurately estimated or effectively unused. In these scenarios, the model can underperform compared to simpler or purely continuous alternatives that do not rely on capturing sudden changes. For applications where jumps are absent or extremely rare, users should first verify the presence (or likelihood) of discontinuities in their dataset before adopting our framework. Additionally, one potential extension is to design an adaptive mechanism that can automatically deactivate or regularize the jump component when the data do not exhibit significant jump behavior, thereby reducing unnecessary complexity and improving general performance on smooth series.

### D.5 Vanilla Euler Solver

---

**Algorithm 3** Vanilla Euler-Maruyama Method

---

**Require:** Total solver steps $M$
1: $\boldsymbol{C} \sim \mathcal{D}_{\text{test}}$, with $\boldsymbol{C} = [S_{-T_p:0}, C]$
2: $\{\mu_\tau, \sigma_\tau, \lambda_\tau, \nu_\tau, \gamma_\tau\}_{\tau=1}^{T_f} \leftarrow f_\theta(\boldsymbol{C})$
3: $\Delta \leftarrow \frac{T_f}{M}$          $\triangleright$ Solver time-step
4: **for** $i = 0, \cdots, M-1$ **do**
5:      $t_i \leftarrow i\Delta, t_{i+1} \leftarrow (i+1)\Delta, \rho_{t_i} \leftarrow \lfloor t_i \rfloor + 1$
6:      $\alpha_i \leftarrow (\mu_{\rho_{t_i}} - \lambda_{\rho_{t_i}} k_{\rho_{t_i}} - \sigma_{\rho_{t_i}}^2/2)\Delta$          $\triangleright$ Drift
7:      $\beta_i \leftarrow \sigma_{\rho_{t_i}}\sqrt{\Delta}z_1$, with $z_1 \sim \mathcal{N}(0,1)$          $\triangleright$ Diffusion
8:      $\zeta_i \leftarrow \kappa\nu_{\rho_{t_i}} + \sqrt{\kappa}\gamma_{\rho_{t_i}}z_2$
     with $\kappa \sim \text{Pois}(\lambda_{\rho_{t_i}}\Delta), z_2 \sim \mathcal{N}(0,1)$          $\triangleright$ Jump
9:      $\ln \bar{S}_{t_{i+1}} \leftarrow \ln \bar{S}_{t_i} + \alpha_i + \beta_i + \zeta_i$
10: **return** $\{\bar{S}_{t_i}\}_{i=1}^{M}$

---

We present the standard Euler–Maruyama solver in Alg. 3, which is used in the ablation study for comparison with our restarted Euler solver.

# E  Impact Statement

This paper introduces Neural MJD, a learning-based time series modeling framework that integrates principled jump-diffusion-based SDE techniques. Our approach effectively captures volatile dynamics, particularly sudden discontinuous jumps that govern temporal data, making it broadly applicable to business analytics, financial modeling, network analysis, and climate simulation. While highly useful for forecasting, we acknowledge potential ethical concerns, including fairness and unintended biases in data or applications. We emphasize responsible deployment and continuous evaluation to mitigate inequalities and risks.

