# OpenReview forum: "Neural MJD: Neural Non-Stationary Merton Jump Diffusion for Time Series Prediction"
_NeurIPS.cc/2025/Conference — NeurIPS 2025 poster_

### Official Review · Reviewer_qPpS · 2025-06-19

**Clarity:** 3
**Significance:** 2
**Originality:** 3
**Rating:** 4
**Confidence:** 4

**Summary:**

In this paper, non-stationary Merton jump diffusion is used to model and predict time series. The model parameters, as five functions of time, are ''predicted'' by a neural network. The practical issues in both training phase (evaluating likelihood) and inference phase (time discretization) are well addressed. Experiments on synthetic and real-word data show that the proposed method has promising performance (low prediction error).

**Questions:**

See ''Strengths And Weaknesses''.

**Ethical Concerns:**

["NO or VERY MINOR ethics concerns only"]

**Final Justification:**

In my review, my main concern is about whether the learning task is well-defined in the sense of standard statistical inference.
After discussion, the authors clarify that they are working in a different amortized setting. After making the difference clear in the revision, I think it will be a good paper.

**Limitations:**

See ''Strengths And Weaknesses''.

**Quality:**

2

**Strengths And Weaknesses:**

The paper is easy to follow and its structure is complete (baseline, proposed model, addressing implementation challenges, performance evaluation). My concern is mainly about the proposed model itself.

At the beginning of Section 4.1, when the non-stationary MJD is introduced, there is no structural assumption on the five model parameters $\{\mu_t,\sigma_t,\lambda_t,\gamma_t,\nu_t\}$, so they can literally be any deterministic functions of $t$ (integrability is mentioned later to make the solution well-defined and continuity is provided by the neural network expression). With this nearly arbitrary flexibility, ''fitting data to non-stationary MJD'' does not seem to be a well-defined task. It is like ''fitting a sequence of arrivals to a non-stationary Poisson process with arbitrary $\lambda_t$'', which leads to sheer overfitting.

The expression (8) $\mu_t,\sigma_t,\lambda_t,\gamma_t,\nu_t=f_\theta(S_0,S_{-1},...,S_{-T_p},C,t)$ may also be problematic. The deterministic future parameters ($\mu_t,\sigma_t,\lambda_t,\gamma_t,\nu_t$) is expressed as a deterministic function ($f_\theta$) of past sample path ($S_0,S_{-1},...,S_{-T_p}$), which is a sequence of random variables. This violates the internal logic of the SDE (unless those parameters are some adaptive processes, not mentioned in the paper), especially when $\mu_t,\sigma_t,\lambda_t,\gamma_t,\nu_t$ are generated from $S_0,S_{-1},...,S_{-T_p}$ in a sliding-window manner as in Section 5.1. Defining instead predicting $\mu_t,\sigma_t,\lambda_t,\gamma_t,\nu_t$ by (8) (in a sliding-window manner) makes more sense to me, but that will lead to a much more complicated SDE (7) + (8) than non-stationary MJD with deterministic $\mu_t,\sigma_t,\lambda_t,\gamma_t,\nu_t$, so all  equations after (8) no longer holds.

''Open access to data and code, Answer: Yes, Justification: We will release our code once the paper is accepted.'' Why?

---

> ### Author Rebuttal · Authors · 2025-07-31
>
> We appreciate the reviewer’s thorough evaluation. We believe that the concerns stem from some technical misunderstandings of our approach, and we value the opportunity to address each point below.
>
> ## Model discussion
> > At the beginning of Section 4.1, when the non-stationary MJD is introduced, there is no structural assumption on the five model parameters, so they can literally be any deterministic functions of $t$ (integrability is mentioned later to make the solution well-defined and continuity is provided by the neural network expression). With this nearly arbitrary flexibility, ''fitting data to non-stationary MJD'' does not seem to be a well-defined task. It is like ''fitting a sequence of arrivals to a non-stationary Poisson process with arbitrary $\lambda_t$'', which leads to sheer overfitting.
>
> **Well-defined task concern**
>
> We respectfully **disagree** with the statement that fitting a non-stationary MJD is not a well-defined task.
>
> Previous models, such as Neural Jump Stochastic Differential Equations [1] and Neural Jump-Diffusion Temporal Point Processes (NJDTPP) [2], adopt highly similar processes to learn SDEs from data using neural networks.
> We find the claim of the task not being "well-defined" to be a vague assessment that contradicts established findings in the research community.
>
> Specifically, in our implementation, we enforce that all learned parameters satisfy the classical MJD requirements. For example, jump intensities and variances are constrained to remain non-negative. This effectively regularizes the learning of SDE parameters.
>
>
> **Overfitting concern**
>
> We believe the reviewer’s comparison to "fitting a sequence of arrivals to a non-stationary Poisson process with arbitrary $\lambda_t$" may stem from a misunderstanding. Rather than fitting each series independently (e.g., as done in ARIMA), we train a single neural network using all sequences in the training data. This approach encourages the model to learn shared non-stationary dynamics, preventing it from merely memorizing individual trajectories.
>
> In practice, we use non-overlapping train/validation/test splits and select model checkpoints based solely on validation performance to avoid overfitting. The experimental results on the unseen testing set clearly show no sign of what is referred to as "sheer overfitting."
>
>
> > The expression (8) $\mu_t,\sigma_t,\lambda_t,\gamma_t,\nu_t=f_\theta(S_0,S_{-1},...,S_{-T_p},C,t)$ may also be problematic. The deterministic future parameters $\mu_t,\sigma_t,\lambda_t,\gamma_t,\nu_t$ is expressed as a deterministic function ($f_\theta$) of past sample path ($S_0,S_{-1},...,S_{-T_p}$), which is a sequence of random variables. This violates the internal logic of the SDE (unless those parameters are some adaptive processes, not mentioned in the paper), especially when $\mu_t,\sigma_t,\lambda_t,\gamma_t,\nu_t$ are generated from $S_0,S_{-1},...,S_{-T_p}$ in a sliding-window manner as in Section 5.1. Defining instead predicting $\mu_t,\sigma_t,\lambda_t,\gamma_t,\nu_t$ by (8) (in a sliding-window manner) makes more sense to me, but that will lead to a much more complicated SDE (7) + (8) than non-stationary MJD with deterministic $\mu_t,\sigma_t,\lambda_t,\gamma_t,\nu_t$, so all equations after (8) no longer holds.
>
> **SDE logic concern**
>
> We respectfully **disagree** that our method violates the internal logic of the SDE.
>
> To avoid confusion, we would like to emphasize that we do not fit jump-diffusion parameters by directly calibrating them on historical data. Instead, past observations $S_{-1},...,S_{-T_p}$ serve only as inputs to a shared neural network. In other words, they are not part of the SDE state variables to be modeled. They can be incorporated into the context information, denoted by $\mathcal{C}$, in our setup.
>
> To be specific, in our model, the SDE process at time $t$ depends only on information available up to time $t$, with no "peeking" into the future. The model has no exposure to future data before making predictions, which is standard practice in deep learning. All the SDE parameters are predicted based on past observations using neural networks once and for all.
>
> We noted some ambiguity in the comment. Our model was described as "sliding-window," yet the reviewer also suggested that "sliding-window makes more sense." We interpret this as a possible preference for an autoregressive approach (please kindly clarify if we have misunderstood the review).
>
> That said, we would like to clarify that our model does not operate in a sliding-window manner. Instead, it adopts a feedforward design that (1) predicts the SDE parameters for the target horizon and then (2) performs numerical simulation to generate actual forecasts. This architecture allows for a single forward pass to predict SDE parameters across all future time steps under consideration, significantly improving inference efficiency by using neural network only once.
>
> ## Open source
> >''Open access to data and code, Answer: Yes, Justification: We will release our code once the paper is accepted.'' Why?
>
> We promise to release our full codebase and data upon acceptance since we would like researchers to be able to reproduce our work. We will release our full codebase and data upon acceptance, as we expect to include additional experiments and corresponding code during the rebuttal phase.
>
>
> ## Concluding remarks
>
> We thank the reviewer again for their valuable feedback. We hope that our rebuttal addresses their questions and concerns, and we kindly ask the reviewer to consider a fresher evaluation of our paper if the reviewer is satisfied with our responses. We are also more than happy to answer any further questions that arise.
>
> ## Reference
>
> [1] Jia, Junteng, and Austin R. Benson. "Neural jump stochastic differential equations." Advances in Neural Information Processing Systems 32 (2019).
>
> [2] Zhang, Shuai, et al. "Neural jump-diffusion temporal point processes." Forty-first International Conference on Machine Learning. 2024.

---

> > ### Comment · Reviewer_qPpS · 2025-08-01
> > **Discussion**
> >
> > Thanks for the rebuttal. In my review, the main concern (well-defined-ness) is about how this deep learning framework fits into the "standard" paradigm of statistical inference for stochastic processes (whether it is well-defined in this sense). If the two papers mentioned adopt a similar setting to the current paper, then my concern extends to those two. I think I need further clarification. To make it easier, let's focus on fitting non-stationary Poisson process (characterized by a deterministic rate function $\lambda(t)\geq0$), which is a special case of the proposed model.
> >
> > Let the dataset contain two paths, one jumps at time 1 and 3, the other jumps at time 2 and 4 (horizon 10). To fit the data, from a model family $\Lambda$, we need to find a model $\lambda_*$ that explains the data the best (e.g., maximum likelihood estimator).
> >
> > If $\Lambda$ is the family of constant functions (stationary Poisson), then the MLE $\lambda_*\equiv 0.2$ to see two arrivals in [0,10] on average. If $\Lambda$ contains all positive functions, then $\lambda_*$ becomes a zero function with 4 bumps at 1,2,3,4, each has width $\epsilon$ and height $1/2\epsilon$ where $\epsilon\downarrow0$ (overfitting).
> >
> > What is the $\Lambda$ in this paper? (It should be between the above two extremes, right?) Is it described in the paper?

---

> ### Author Response · Authors · 2025-08-04
> **Reply to Reviewer's Discussion**
>
> We thank the reviewer for engaging in the discussion.
>
> ### Amortized vs. Standard Statistical Inference
>
> First, we would like to clarify that our goal is not to merely "fit to the standard paradigm of statistical inference." Instead, we adopt the **amortized inference** principle by training a shared neural network that maps context features to SDE parameters. This neural network is trained across a large, diverse dataset capturing a wide range of contextual variability. At test time, the pre-trained network takes in unseen context features and outputs parameter estimates in a single forward pass, without any additional optimization.
>
> ### Conditional vs. Unconditional Modelling
>
> Second, our model is trained using a **conditional MLE** objective. It conditions on a past window of time-series observations, and optionally on static features (e.g., store attributes in business revenue prediction). This stands in contrast to the reviewer's toy scenario, which assumes no conditional input. In real-world datasets such as stock prices or store revenues, each sample typically has non-overlapping context features; identical context configurations are not to be found in our actual datasets, excluding one-to-many ambiguity in implementation.
>
> ### Overfitting
>
> Regarding the reviewer's overfitting concern in the non-stationary Poisson process example, we respectfully disagree with the conclusion. In the second case, involving a model family with all-positive $\Lambda$ functions, the estimated $\lambda_\star$ achieves the lowest training error by capturing four bumps. However, low training error (i.e., good fit or perfect fit) does not, on its own, imply overfitting (i.e., poor generalization to unseen data). In the toy example provided, there is no quantitative evaluation or empirical evidence of generalization performance on held-out data. While we acknowledge that more flexible model families can achieve lower training error, this does not necessarily lead to poor generalization—it must be assessed through explicit experimental validation.
>
> On the contrary, prior research shows that perfect fitting does not necessarily hinder generalization for neural models. In [1], the authors demonstrated the phenomenon of *benign overfitting*, showing that least-squares interpolators can generalize well despite zero training error in certain high-dimensional linear models. In [2], it was shown that large neural networks can memorize random labels while still generalizing effectively on real data. Moreover, we observe no signs of overfitting in our experiments, as our model consistently achieves competitive quantitative performance compared to various baselines.
>
> Overall, we thank the reviewer for engaging in the discussion and hope our rebuttal has addressed the concerns raised. We remain open to further dialogue.
>
> Reference:\
> [1] Bartlett, Peter L., et al. "Benign overfitting in linear regression." Proceedings of the National Academy of Sciences. 2020.
>
> [2] Zhang, Chiyuan, et al. "Understanding deep learning requires rethinking generalization." International Conference on Learning Representations. 2017.

---

> > ### Comment · Reviewer_qPpS · 2025-08-04
> > **Thanks**
> >
> > Thanks for the clarification. Now it is clear that there are two very different worlds. In the revision, the authors may consider clearly explaining their logical/methodological/foundational difference after introducing the SDE, i.e., expanding the short paragraph around (8), so that readers with the "standard statistical inference" mindset can switch gear to follow the rest of the paper (e.g., to understand (8) itself).
> >
> > I will raise my score to 4.

---

> > > ### Author Response · Authors · 2025-08-05
> > >
> > > Dear Reviewer qPpS,
> > >
> > > We sincerely appreciate the time and effort you have dedicated to reviewing our work and participating in the discussion. Thank you for your insightful comments and questions, which have greatly helped us clarify and strengthen our submission. We will incorporate the key points from the discussion as we continue to refine the manuscript during the revision.
> > >
> > > Best regards,\
> > > The Authors

---

### Official Review · Reviewer_scGv · 2025-06-30

**Clarity:** 4
**Significance:** 3
**Originality:** 4
**Rating:** 5
**Confidence:** 3

**Summary:**

This paper introduces time-inhomogeneous Merton jump diffusion models for stochastic time-series prediction. The authors introduce a truncated likelihood that makes learning the model practical, and also a 'restarting' Euler-Maruyama solver that has a tighter error bound. The model is evaluated on a number of synthetic and real-world datasets, against a large number of baseline models and performs well.

**Questions:**

Q1: The first experiment is on synthetic data, i.e., you know the *true* parameters of the MJD process that generates the data. Can you please show how the estimated parameters using your method compare to the true parameters?

Q2: You sample 10 times to express mean metrics. Can you please also provide confidence intervals on these metrics? E.g. using t-test statistics as is done in Bartosh SDE Matching (2025) and earlier works on latent SDEs.

Q3: You restart the EM solver at each observation. Do you think this has a negative effect on learning long-term dependencies? I mean, a change of the parameters at some time t will not have an effect on the dynamics after the next S_\tau is crossed?

Q4: I'm a bit surprised you needed to add the 'regularization' term. It resembles a Gaussian likelihood on the measurements (L2 term). Can you provide me more intuition on why (or why not) your likelihood does not already contains a similar effect?

Q5: The related work on jump processes seems to focus on the financial application domain. But inference of jump processes has been investigated in machine learning, how does your work relate to these 'older' works? e.g.
https://academic.oup.com/jrsssb/article-abstract/67/3/395/7109469
https://proceedings.neurips.cc/paper/2007/hash/735b90b4568125ed6c3f678819b6e058-Abstract.html


Minor questions or remarks:
- line 166: There seems to be an unnecessary space at the start of this line.
- line 178, this means you always discretize with time-step equal to the time unit?
- line 233 parallel

**Ethical Concerns:**

["NO or VERY MINOR ethics concerns only"]

**Final Justification:**

Ok, thanks for your answers and the extra work during rebuttal. I retain my score.

**Limitations:**

yes

**Paper Formatting Concerns:**

No concerns.

**Quality:**

3

**Strengths And Weaknesses:**

Strengths:
- The paper is very well written. The work is clearly embedded and motivated in the context of (primarily) financial modelling. Specific concepts are explained with high clarity, this is very much appreciated. As well the clear citations to e.g. specific book chapters allows the reader to grasp more context.
- The method is well explained. While the model might be rather specific, and not e.g. a general SDE driven by Lévy, I would like to see this as a plausible and rigorous step forwards for the research community.
- Modelling stochastic jump processes is a difficult problem, which is very relevant to the machine learning community.
- The experiments are convincing and extensive.

Weaknesses:
- Experiments are done by sampling 10 times from the models. Results don't show confidence intervals, and sometimes other models (such as flow matching) are very close the the author's model, but not indicated in bold. This is a bit misleading. I would like to see an evaluation where, if the best models are performing similar (statistically significantly), they are both in bold.

---

> ### Author Rebuttal · Authors · 2025-07-31
>
> We sincerely thank the reviewer for their time and effort in evaluating our work. We greatly appreciate their recognition of our model formulation and the application of MJD SDE modeling to time-series analysis. Below, we address the key points raised in the review.
>
> ## Confidence interval
> > I would like to see an evaluation where, if the best models are performing similar (statistically significantly), they are both in bold.
>
>
> > You sample 10 times to express mean metrics. Can you please also provide confidence intervals on these metrics? E.g. using t-test statistics as is done in Bartosh SDE Matching (2025) and earlier works on latent SDEs.
>
> Thank you for your valuable suggestions. We provide additional experimental results on the SafeGraph & Advan business analytics dataset, where we compute the confidence interval by calculating the standard deviation across multiple runs. The preliminary experiments are as follows:
>
>
> |Model|MAE $\downarrow$|R² $\uparrow$|
> |----------|------|--------|
> |DDPM| 68.5 $\pm$ 3.1 |0.501 $\pm$ 0.023|
> |FM | 54.5 $\pm$  2.4| 0.540 $\pm$ 0.020|
> |**Ours**|54.1 $\pm$ 1.8|0.549 $\pm$ 0.018|
>
> Due to the time constraints of the rebuttal period, we will compute the t-test statistics as suggested in future work and include these metrics for all stochastic methods in additional experiments.
>
>
> ## Synthetic dataset
> > The first experiment is on synthetic data, i.e., you know the true parameters of the MJD process that generates the data. Can you please show how the estimated parameters using your method compare to the true parameters?
>
>
> Thank you for your feedback.
> We show the estimation parameter differences for the $\mu$ and $\sigma$ parameters using our Neural BS and full Neural MJD models. The results are averaged over time steps to obtain aggregated MAE results.
>
> | Model      | $\mu$ error | $\sigma$ error |
> |------------|--------|----------|
> | Neural BS  | 0.15   | 0.21     |
> | Neural MJD | 0.11   | 0.19     |
>
> As we can see, the Neural MJD model performs slightly better, likely due to its ability to model jumps during the learning process, whereas the Neural BS model only captures the SDE without jumps.
> For the other jump-related parameters, we don’t observe very close estimations from the Neural MJD. We hypothesize that different combinations of jump parameters may yield similar outcomes in the likelihood loss training.
> We plan to devote more effort to exploring this aspect further in order to enhance model interpretability. Thank you again for your insightful comment.
>
>
> ## Restart mechanism
> > You restart the EM solver at each observation. Do you think this has a negative effect on learning long-term dependencies? I mean, a change of the parameters at some time t will not have an effect on the dynamics after the next S_\tau is crossed?
>
> We would like to respectfully point out that the restart mechanism is based on the conditional mean, which is given by:
> $$
>     \mathbb{E}\left[S_{t} \vert \mathcal{C} \right] = S_0 \exp(\sum_{j=1}^{\rho_t-1} \mu_{j} + (t-\rho_t+1) \mu_{\rho_t}).
> $$
>
> As such, the SDE parameters $\mu_t$ still implicitly capture the long-term dependencies learned by the neural network.
>
> We agree that changes in other parameters have a lesser impact on the SDE dynamics once $S_\tau$ is crossed. This observation aligns with our design motivation, which aims to reduce simulation variance in the long term to achieve improved empirical performance.
>
> ## Regularization loss
>
> >I'm a bit surprised you needed to add the 'regularization' term. It resembles a Gaussian likelihood on the measurements (L2 term). Can you provide me more intuition on why (or why not) your likelihood does not already contains a similar effect?
>
> Thanks for your comment.
> The extra regularization loss is added to address the practical issues associated with the likelihood-based learning objective.
> The intuition behind this is that a low likelihood loss does not necessarily guarantee that the conditional mean (as represented by the regularization loss) is well learned.
> Due to the jump components of the SDEs, we conjecture that there may be local minima that result in a relatively low likelihood loss, but where the conditional mean can still vary significantly.
> Since our restart solver also uses the conditional mean, we find that adding this term generally enhances performance by constraining the optimization space when using likelihood-based loss.
>
>
> ## Related work
> > The related work on jump processes seems to focus on the financial application domain. But inference of jump processes has been investigated in machine learning, how does your work relate to these 'older' works?
>
> We thank the reviewer for highlighting these foundational works on jump-process inference. It is an insightful suggestion to include a comparison with classical inference methods for jump processes.
>
> While these methods share the goal of estimating non-stationary parameters from discrete observations, our approach differs in three key respects:
>
> 1. Inputs vs. Calibration: We do not fit SDE parameters directly to past data using Maximum Likelihood Estimation (MLE). Instead, we treat historical observations and additional information as inputs to the neural network. The likelihood serves as a training loss to optimize forecast accuracy, rather than as a per-series estimator of past states.
>
> 2. Classical Methods: Classical Expectation-Maximization (EM) or Markov Chain Monte Carlo (MCMC)-based methods estimate parameters for each individual sequence. In contrast, our single network is trained once over all series, capturing shared dynamics and generalizing across assets or nodes.
>
> 3. Closed-Form Requirement: Classical methods do not require a closed-form transition density, while our method does. This limitation provides a promising direction for future work, where the closed-form constraint could potentially be relaxed.
>
> These "older" works provide a strong statistical foundation and could inspire hybrid methods. We will incorporate a discussion of their contributions and distinctions in the Related Work section of the revision.
>
> Additionally, we find that these works offer valuable insights into the challenges and non-uniqueness of estimating non-stationary event rates from discrete observations (which relates to the previous question).
>
>
>
> ## Writing
> > line 178, this means you always discretize with time-step equal to the time unit?
>
> In practice, we hold the parameters fixed during the time unit (1 in this case), but during this time unit, you can choose any smaller $\delta$ to reduce discretization error.
>
>
> We thank the reviewer for pointing out the typos and formatting issues. We will address these in the revision.
>
> ## Concluding Remarks
>
> We sincerely thank the reviewer once again for their valuable feedback. We hope our responses have satisfactorily addressed their questions and concerns. We remain fully available and happy to answer any further questions that may arise.

---

> > ### Comment · Reviewer_scGv · 2025-08-04
> >
> > Ok, thanks for your answers and the extra work during rebuttal. I retain my score.

---

### Official Review · Reviewer_J6Nv · 2025-07-01

**Clarity:** 4
**Significance:** 3
**Originality:** 3
**Rating:** 5
**Confidence:** 3

**Summary:**

This work introduces Neural MJD, a non-stationary Merton Jump Diffusion model parameterized by a series of neural networks, which captures a wide variety of stochastic processes particularly in the world of quantitative finance. The authors additionally propose a restart inference method, based on the celebrated Euler-Maruyama method which achieves exponentially tighter weak error, and demonstrate the efficacy of their methods on a swath of synthetic and real-world bench-marks.

**Questions:**

* What are the implications of this work for modeling option prices and more generally volatility?
* In the proof of Theorem 4.1, what are the precise assumptions necessary for this theorem to hold?
* Does the method obtain improved results when the discretization is performed on smaller intervals?

**Ethical Concerns:**

["NO or VERY MINOR ethics concerns only"]

**Final Justification:**

This work introduces a novel model whose experimental results surpass the best known alternatives across a range of benchmarks relevant to finance, economics, and other domains. Towards tractable learning, the authors present a likelihood truncation mechanism and provide theoretical error bounds for this approximation. The effectiveness of their methods is demonstrated on a broad set of synthetic and real-world benchmarks.

My primary concern with the work was with the S & P data experiment, worrying that a year and a half of data with relatively low market volatility may not be general enough to prove the efficacy of their methods. However, the authors have since extended the time frame and shown that their methods remain effective under the more challenging conditions. Moreover, the authors also answered my questions, revealing insights on the nature of the discretization interval. They also included additional experiments addressing concerns raised by other reviewers.

For these reasons, I maintain my positive evaluation and recommend a score of 5 (Accept).

**Limitations:**

Yes.

**Quality:**

3

**Strengths And Weaknesses:**

> Strengths:
>  * The exposition in the text is clear and the motivation is well defined.
>  * The method proposed in the paper is novel, and subsumes the important Black-Scholes and Merton Jump Diffusion models to account for non-stationarity by way of encoding the parameters in a time-varying transformer architecture.
>   * The model derivation in Section 3 as well as the deferred proofs in the appendix are well-written and provide clear motivation
>  * The experiments are run on a swath of synthetic and real-world tests, including modeling the S & P index, demonstrating the efficacy of their novel method.
>  * The authors give many helpful figures to explain certain phenomena.

> Weaknesses:
> * The S & P experiment is run on a limited set of data (Jan 2016 - Apr 2017), which had limited volatility (evidenced by the VIX index during this time), it is unclear whether this method still outperforms given the increased volatility of the past 5 years.
> * Additional discussion on the relationship between this work and other works relating neural learning in non-stationary time series would be appreciated.

---

> ### Author Rebuttal · Authors · 2025-07-31
>
> We would like to express our sincere gratitude to the reviewer for their time and effort in evaluating our work. We are grateful for their acknowledgment of the novelty in our model formulation and the application of MJD SDE modeling to time-series analysis. Below, we respond to the key points raised in the review.
>
>
> ## Additional experiments
> >The S & P experiment is run on a limited set of data (Jan 2016 - Apr 2017), which had limited volatility (evidenced by the VIX index during this time), it is unclear whether this method still outperforms given the increased volatility of the past 5 years.
>
> Thank you for your valuable feedback.
> We have extended the S&P 500 dataset to cover the period from January 2014 to December 2017, using the data from 2017 for performance evaluation. The preliminary experimental results are as follows:
>
> |Model|MAE $\downarrow$|R² $\uparrow$|
> |----------|------|--------|
> |ARIMA | 65.2 | -0.892 |
> |XGBoost| 56.4 |0.150|
> |MLP| 51.2 | 0.161 |
> |**Ours**|**39.7**|**0.389**|
>
> In the extended dataset, our method still outperforms the baselines.
> In future work, we plan to extend our experiments to cover the past five years comprehensively, alongside additional baselines.
> We will also analyze the empirical effects of long-term forecasting and data volatility on model performance.
>
>
> ## Discussion on related work
> >Additional discussion on the relationship between this work and other works relating neural learning in non-stationary time series would be appreciated.
>
> We thank the reviewer for the helpful suggestion to clarify how our work relates to prior research on non-stationary time series. We will include this clarification in our revision to prevent any ambiguity.
>
> In previous studies, "non-stationarity" typically refers to distributional shifts in the data over time. These works focus on mitigating such shifts using techniques like input-level normalization (e.g., DAIN [1], ST-norm [2], RevIN [3]), domain adaptation (e.g., DDG-DA [4]), or de-stationary attention mechanisms (e.g., Non-stationary Transformers [5]).
>
> In contrast, our notion of non-stationarity centers on modeling a Merton jump diffusion (MJD) process with parameters that evolve over time. While our model is not explicitly designed to handle distributional shifts, it predicts time-varying MJD parameters through a closed-form solution, which naturally leads to a changing predictive distribution as parameters evolve.
>
>
> ## Model implications
> > What are the implications of this work for modeling option prices and more generally volatility?
>
> Thank you for your feedback. We would like to highlight the following advantages of our approach:
>
> 1. Reduced calibration effort: Classical option-pricing methods require significant effort to fit a time-inhomogeneous Merton jump-diffusion model to market or historical data. In contrast, Neural MJD treats past returns as conditioning inputs rather than direct calibration targets and directly outputs the drift, diffusion, and jump-intensity parameters for the future time horizon. This approach eliminates the need of time-consuming calibration due to fast neural network inference.
>
> 2. Handling exogenous information: Most classical modeling methods either ignore exogenous drivers or can only handle numerical features. Our method, however, is capable of incorporating diverse types of data as inputs or conditioning variables, such as static features (e.g., parking lot sizes) in the SafeGraph and Advan datasets used for business revenue prediction.
>
>
> 3. Potential extension to other SDE models: A similar neural parameterization technique can be extended to other SDE dynamics for more domain-specific applications, allowing for better capture of volatility. The training and inference techniques introduced in the paper would be beneficial for future model development.
>
>
> ## Theorem proof
>
> > In the proof of Theorem 4.1, what are the precise assumptions necessary for this theorem to hold?
>
> Theorem 4.1 (the truncation-error bound for the non-stationary MJD) is proven under the following assumptions:
>
>
> 1. We choose $\delta$ small enough that $t$ and $t+\delta$ are under the same set of parameters. We have this assumption simply to ease the proof of the theorem.
> 2. Other standard Poisson/MJD assumptions:
>     a. The jump sizes in Poisson process are i.i.d.
>     b. Conditional on $\Delta N$, $\ln S_{t+\delta}$ is Gaussian.
>
> We will clarify the assumptions used in the proof of the theorem, and we thank the reviewer for bringing this up.
>
>
> ## Discretization interval
> > Does the method obtain improved results when the discretization is performed on smaller intervals?
>
> Theoretically, using a finer discretization reduces both the Euler integration error and the jump-truncation bias per interval. Empirically, we find that using an interval that is small enough (such as 0.1 or 0.2) leads to good performance.
> We conduct the comparison on the S&P 500 dataset as follows.
> Please note that the interval is normalized relative to the time difference between two times of interest (which is 1 in our notation).
>
>
> | Intervals | 0.05 | 0.1  | 0.2  | 0.5  | 1.0  |
> |-----------|-------|-------|-------|-------|-------|
> | Mean MAE $\downarrow$       | 17.1  | **16.8**  | 17.1  | 35.2  | 43.2  |
>
>
> ## Concluding Remarks
>
> We sincerely thank the reviewer once again for their valuable feedback. We hope our responses have satisfactorily addressed their questions and concerns. We remain fully available and happy to answer any further questions that may arise.
>
>
> ## Reference
>
> [1] Passalis, Nikolaos, et al. "Deep adaptive input normalization for time series forecasting." IEEE transactions on neural networks and learning systems 31.9 (2019): 3760-3765.
>
> [2] Deng, Jinliang, et al. "St-norm: Spatial and temporal normalization for multi-variate time series forecasting." Proceedings of the 27th ACM SIGKDD conference on knowledge discovery & data mining. 2021.
>
> [3] Kim, Taesung, et al. "Reversible instance normalization for accurate time-series forecasting against distribution shift." International conference on learning representations. 2021.
>
> [4] Li, Wendi, et al. "Ddg-da: Data distribution generation for predictable concept drift adaptation." Proceedings of the AAAI Conference on Artificial Intelligence. Vol. 36. No. 4. 2022.
>
> [5] Liu, Yong, et al. "Non-stationary transformers: Exploring the stationarity in time series forecasting." Advances in neural information processing systems 35 (2022): 9881-9893.

---

> > ### Comment · Reviewer_J6Nv · 2025-08-04
> >
> > Thank you for the additional experiments and clarification of the assumptions required for Theorem 4.1. I maintain my positive evaluation.

---

### Official Review · Reviewer_PuBS · 2025-07-02

**Clarity:** 3
**Significance:** 2
**Originality:** 2
**Rating:** 5
**Confidence:** 3

**Summary:**

This paper presents the Neural MJD, a transposition of ideas from neural ODEs into a specific family of differential equations optimized to handle data with jumps. A collection of efficiency-improving suggestions are made. These are then benchmarked on a small-scale synthetic experiment, and two financial datasets. The method appears to perform well.

**Questions:**

1. What is the “MLP” baseline.  Would like to see how best-in-class forecasting algorithms such as N-BEATS perform (even though they can’t handle irregularly spaced data).
2. Are the time series used univariate or multivariate?  I can’t quite tell from the description in 5.2. Then how do you use some baselines like ARIMA that don’t handle vector valued data? (Or are you actually use VARIMA models?)
3. (cf. line 208) Something I don’t quite understand is how the model produces inferences for any t with a single evaluation?  This means that it is basically a feed-forward network that outputs the effective Euler derivative of the state (multiplied by the time interval and added to the previous state results in the new estimate).
    - Similarly, this is quite a departure from the neural differential equations way of operating, right?  To the point that it isn’t really a neural ODE that _requires_ integrating; or am I missing something…?
4. (cf. Section 4.3) Trying to square this with Algorithm 2, why do you need to do the integration steps at all?  Why not just evaluate the exact target value and their variances at the times of interest?  I don’t love the introduction of these restart points where the prediction radically changes.  I can see that it improves performance, but it feels messy and unfinished. I would be more interested if you then fitted a GP or did smoothing over it to at least get a smooth estimate of the mean and variance which is constrained by these “gold-standard” evaluations at the restart points.

**Ethical Concerns:**

["NO or VERY MINOR ethics concerns only"]

**Final Justification:**

Thank you to the authors for their response.  As far as I am concerned: the original paper seemed correct and it seemed to work well within the applied domain.  In the rebuttal phase, the authors have made some attempt to expand the domains and to test more baselines.  Based on this, I find no reasonable cause from my own review to not upgrade my score (although I do note not being able to see the updated manuscript/results/baselines/justification tempers my confidence in this somewhat).

I am also waiting to see how the discussion with qPpS plays out.  I don't fully understand the core critique: it seems like the discussion is about the ability of the model to overfit, which is valid, but is also a relatively omnipresent criticism that can be resolved (regularization etc).  If the model performs well in the domains it is tested on, then I find it difficult to add credence to the reviewers line of questioning.  I will track the discussion and will upgrade my final score if there is no substantial flaws found.

**Limitations:**

No societal impacts.

Main technical limitations are in reach and strength of empirical validation.

**Paper Formatting Concerns:**

None.

**Quality:**

2

**Strengths And Weaknesses:**

## Summary
I think this paper is an interesting and very detailed assessment of extending themes from neural differential equations into very specialized model families.  It appears well motivated from a historical and finance perspective, then tying in nicely with some modern themes. I have some reservations really striking at the generality and strength of evaluation. As there is not enough differentiating material across a broad base, I find myself being _indifferent_ to the paper (it seems to work fine under a narrow definition), and find it difficult to campaign ardently for its inclusion (or indeed exclusion).  Since I can see this paper having some impact and is a nice combination of very domain-specific models with very general concepts, I err towards acceptance. I would like to see a revision to this paper that tackles more broad use-cases, compares to more appropriate baselines, and attempts to leverage the underlying structure of the model to unlock new insights.  Good luck.

## Strengths
1.	Modeling data with jumps is a practical tool, since real-world data is often very messy with lots of jumps.
2.	A lot of the paper is well written and clearly expressed despite some pretty intense material (modulo some points of confusion I do have; see Questions).
3.	I think the restarts are an interesting idea.
4.	I like some of the simplifications and modeling assumptions, enabling easier inference.
5.	Seems to perform admirably.
6.	Appreciate the inclusion of runtime comparison.
7.	The authors are very precise in their exposition and should be commended as such.  I did lose the thread a little bit around 4.3, I have detailed some questions below.

## Weaknesses (roughly in order of significance)
1. My main weakness with this paper is that it is presented in a very domain-specific way.  the model is effectively only benchmarked on two univariate real-world prediction tasks under fairly restricted settings, which does temper my enthusiasm for the generality and potential impact of the method. I would love to see it benchmarked on some non-financial timeseries, because I think there could be some real performance benefits in other domains (e.g. vitals monitoring in healthcare).  Similarly, it is a transplantation (if a very thorough and precise transplantation!) of ideas from neural differential equations into another, more specialized family of differential equations.  This again limits the impact and reach to me, without seeing applications against domain-specific baselines in a wider range of tasks.  I think this is a bit of a missed opportunity.  I don’t think this is terminal in my opinion, the paper still clearly has merit, but it does reduce my enthusiasm.
2. I think the baselines are sort of set up to fail. Looking through the baselines in Table 2: ARIMA models don’t capture jumps, and BS and MJD are “static” versions of your models. XGBoost is good but limited, see question below on MLP, and I don’t know much about the GCN or other diffusion baselines, but I doubt they are very well configured for jumps.  Your model basically has heavier tails than those models, and so I would expect it to do fairly well on data with jumps. I would be much more interested in comparisons with state-of-the-art forecasting methods like N-BEATS and N-HITS, or TCN or TiDE.  I think these are much more representative baselines for this sort of data (and should be relatively easy to compare to, packages like DARTS implement them and have stochastic decoding methods).  Until seeing these more even-handed evaluations, my enthusiasm is still tempered.
3. I do not agree with some of the parlance like “inability to explicitly model underlying stochastic processes often limit their generalization to non-stationary data”. Not only is this unsubstantiated as a claim and biases the reader towards your method unfairly, I don’t think it is correct. Maybe a neural network doesn’t explicitly parameterize an SDE, but I don’t see why that would limit generalization (arguably it would increase generalization!).  Then no model can truly expand to non-stationary timeseries, because the timeseries might enter a dynamics space it hasn’t seen before and so is just out-of-distribution and its randomly guessing.  I don’t see why your model is any more or less robust to this. These claims or hints are scattered throughout the paper. I think choosing to model an SDE directly is interesting, but since you do not discuss mechanistic interpretability, causality, or parameter efficiency (which you also lose because of the neural networks), I think the parameterization with an SDE is incidental and doesn’t directly contribute to this – I think the more powerful framework of defining a neural differential equation contributes to this, and so this language should (to my taste) be stripped out.
4. (this is a comparatively minor point.) I would like to see some of the weight of the exposition stripped out. It is quite a “heavy” paper, for what is a relatively simple idea (at least once it has been worked through). I would love to see a revision that added some more examples and exploration of the results (successes and failures!) and demoted some of the derivations to the supplement. I also find Algorithm 1 and 2 quite unhelpful.  I would prefer if they were spaced out and properly commented to explain what each line does, because a lot of the equations don’t actually appear in the main text (e.g. $\zeta$ randomly appears and is not referenced in the rest of the paper?).
5. I’m not quite sure how you’d improve this, but the $N(dt, dy)$ notation I find confusing.  Maybe you could explicitly separate it into two integrals, and also denote that $N$ is actually a function of $t$? (cf. line 150).  This suppression makes it quite hard to keep a track of the dependencies.

---

> ### Author Rebuttal · Authors · 2025-07-31
>
> We sincerely thank the reviewer for their time and effort in evaluating our work. We appreciate their recognition of the novelty of our model formulation and the use of SDE modeling for time-series analysis. Below, we address the main points raised in the review.
>
>
> ## Applicable domain
> > benchmarked on some non-financial timeseries
>
> To further address this concern, beyond the economic-financial time-series data used in the paper, we conducted an additional experiment predicting **daily visit volumes** to business points of interest (POIs) using the SafeGraph and Advan datasets. Our method continues to outperform the baseline models in this setting as well.
>
> |Model|MAE $\downarrow$|R² $\uparrow$|
> |----------|------|--------|
> |ARIMA|58.2|-0.125|
> |XGBoost|33.4|0.189|
> |MLP|30.3|0.225|
> |GCN|31.4|0.219|
> |FM|27.9|0.273|
> |**Ours**|**27.5**|**0.268**|
>
> As future work, we aim to apply our method to more diverse settings and datasets, such as healthcare datasets as suggested by the reviewer.
>
> > more specialized family of differential equations
>
> We respectfully argue that our approach, inspired by Merton jump diffusion SDEs formulations, provides a broader and more flexible framework than simpler models like the Black-Scholes SDE. This allows our model to better capture jump behavior, which standard neuralized SDEs may struggle with. While fully neural formulations (e.g., NeuralCDE, LatentSDE) can theoretically represent a wider range of SDE processes, they do not outperform our approach as seen in the experiments. Empirical results demonstrate that the use of jump diffusion SDEs in our model design leads to improved performance.
>
>
> ## Baseline fairness
> > I would be much more interested in comparisons with state-of-the-art forecasting methods
>
> We conducted further experiments on the S&P 500 dataset and obtained preliminary results for TCN and N-BEATS:
>
> |Model|MAE $\downarrow$|R² $\uparrow$|
> |----------|------|--------|
> |TCN | 47.9 | 0.215 |
> |N-BEATS| 33.5 |0.751|
> |**Ours**|**16.8**|**0.938**|
>
> We plan to include additional experiments comparing with more baselines in the revised version.
>
> > I think the baselines are sort of set up to fail.
>
> We would like to clarify that our experiments were designed for a fair comparison. Some baselines do not explicitly model jumps, motivating our Neural MJD, and the performance comparison supports the proposed model.
> Except for the synthetic data, we do not control jump behavior in real-world datasets. Our goal is to ensure a fair, practical comparison on real data without any jump labels and to highlight the methodological value of modeling potential jumps.
>
> > What is the “MLP” baseline.
>
> The MLP model consists of a stack of linear layers and serves as a simple baseline for time-series forecasting.
>
> > Are the time series used univariate or multivariate?
>
> We use univariate time-series datasets, predicting a scalar at each time step.
>
>
> ## Generalization to non-stationary data
> > I do not agree with some of the parlance like “inability to explicitly model underlying stochastic processes often limit their generalization to non-stationary data”.
>
> We thank the reviewer for their valuable feedback. Our intention was to emphasize that explicitly modeling jump diffusion SDEs offers a useful inductive bias for capturing sudden changes and non-stationary behaviors that may be more challenging to represent with standard neural SDEs. We believe this approach complements the neural differential equation framework by providing additional flexibility. We appreciate the suggestion and will revise the manuscript to ensure accuracy.
>
> ## Neural ODE vs. ours
>
> > how the model produces inferences for any t with a single evaluation?
>
> > why do you need to do the integration steps at all?
>
> We would like to clarify that our model is not a neural ODE and does not require step-by-step integration of a *neural* vector field. Instead, it predicts all SDE parameters—drift, diffusion, and jump intensity—in a single forward pass. The SDE trajectory $S_t$ is then simulated using these predicted coefficients via a numerical method, without further neural network evaluations.
>
> A neural ODE typically learns a vector field:
> $\frac{dS(t)}{dt} = f(S(t), t; \theta),$ and computes $S(T_f) = S(0) + \int_0^{T_f} f(S(t), t; \theta)\,dt$ using an ODE solver, which repeatedly queries the neural network during integration.
>
> In our method, we do not learn a neural vector field. Instead, we predict the full set of Merton jump-diffusion (MJD) parameters: $[\mu_1, \sigma_1, \lambda_1, \gamma_1, \nu_1, \dots] \in \mathbb{R}^{5 \times T_f}.$ Due to the stochastic nature of SDEs, numerical simulation is required to generate actual forecasts. We compute each next state (informally) as:
> $S_{t+\Delta t} = S_t + \mu_t\,\Delta t + \sigma_t\,\Delta W + \text{jump}(\lambda_t, \gamma_t, \nu_t),$
> which involves stochastic integration—but no further neural network inference, as all parameters are predicted once at the start.
>
> In summary, our model generates predictions for the SDE parameters at time $t$, followed by numerical simulations to obtain SDE trajectories for forecasting outcomes.
>
> > Why not just evaluate the exact target value and their variances at the times of interest?
>
> We would like to emphasize that we do not forecast outcomes in a single step. Instead, we sequentially sample each increment across the entire SDE simulation trajectory to obtain stochastic forecasts. Relying solely on the exact expectation and variance would oversimplify the model, ignoring the jump dynamics. While our formulation allows for the computation of mean and variance, these values may deviate from the true behavior, which includes jumps.
>
> > get a smooth estimate
>
> We thank the reviewer for raising this point. In Tables 1–3, the "Mean" metrics represent averages over 10 runs of the SDE simulation, serving as a simple proxy for smoothing. We agree that more advanced approaches, such as GP fitting, are promising and plan to explore them in future work.
>
> ## Writing clarity
>
> Thank you for your helpful comments. We will clarify the derivations and notation in the main paper and present our method more clearly in the revision. We followed the classical literature for the Poisson random measure $N(\mathrm{d}t, \mathrm{d}y)$ and will also include explanations highlighting its meaning and interpretation in practical applications to enhance understanding.
>
> ## Concluding remarks
> We thank the reviewer again for their valuable feedback. We hope that our rebuttal addresses their questions and concerns, and we kindly ask the reviewer to consider a fresher evaluation of our paper if the reviewer is satisfied with our responses. We are also more than happy to answer any further questions that arise.

---

### Note · Authors · 2025-08-14

We sincerely thank all reviewers for their time, constructive feedback, and recognition of our contributions. The suggestions have helped us clarify and strengthen key aspects of our work. Below, we summarize the main points addressed in our rebuttal.

### Methodology Clarification
We addressed Reviewer qPpS's concern about problem formulation by clarifying that our method uses amortized inference to predict non-stationary MJD parameters from past observations and contextual information in a single forward pass. We further detailed the conditional MLE objective used in our model, demonstrating through empirical analysis that it generalizes effectively across diverse datasets. As a result of this discussion, Reviewer qPpS raised their score. We also clarified that our model is not a variant of neural ODEs and explained the differences in detail.


### Broader Applications and Comparisons
Beyond the store revenue, stock price, and synthetic datasets in the submission, we additionally evaluated on the store daily-visit dataset during rebuttal and confirmed performance advantages.
We added comparisons with SOTA forecasting models (TCN, N-BEATS, TiDE, Autoformer), demonstrating Neural MJD's substantial gains, which contributed to Reviewer PuBS upgrading their scores. Extending the S&P 500 experiment to cover more volatile market periods further confirmed consistent superiority over baselines.


### Ablations and Robustness
We clarified that the restart mechanism reduces variance without losing long-term dependencies, and that the regularization term stabilizes likelihood training by constraining the conditional mean. We also validated robustness via reporting confidence intervals in synthetic experiments.

---

### Decision · Program_Chairs · 2025-09-17

**Decision:**

Accept (poster)

**Comment:**

The reviewers agree that this is an interesting and useful paper, bringing ideas from modern ML to more classical time-series modeling. There were some concerns about clarity, evaluations, and somewhat narrow scope (at least in the presentation). None of these were unanimous, and the authors in their rebuttal suggested some remedies. Please take these into account (as well as the overall author-reviewer discussion when preparing the camera-ready version).